# Generalized Eigenvalue Problems with Generative Priors

**Zhaoqiang Liu**        **Wen Li**[*]
University of Electronic Science and Technology of China
{zqliu12, liwenbnu}@gmail.com

**Junren Chen**
University of Hong Kong
chenjr58@connect.hku.hk

## Abstract

Generalized eigenvalue problems (GEPs) find applications in various fields of science and engineering. For example, principal component analysis, Fisher's discriminant analysis, and canonical correlation analysis are specific instances of GEPs and are widely used in statistical data processing. In this work, we study GEPs under generative priors, assuming that the underlying leading generalized eigenvector lies within the range of a Lipschitz continuous generative model. Under appropriate conditions, we show that any optimal solution to the corresponding optimization problems attains the optimal statistical rate. Moreover, from a computational perspective, we propose an iterative algorithm called the Projected Rayleigh Flow Method (PRFM) to approximate the optimal solution. We theoretically demonstrate that under suitable assumptions, PRFM converges linearly to an estimated vector that achieves the optimal statistical rate. Numerical results are provided to demonstrate the effectiveness of the proposed method.

## 1 Introduction

The generalized eigenvalue problem (GEP) plays an important role in various fields of science and engineering [72, 70, 23, 21]. For instance, it underpins numerous machine learning and statistical methods, such as principal component analysis (PCA), Fisher's discriminant analysis (FDA), and canonical correlation analysis (CCA) [85, 37, 61, 5, 28, 22, 48, 53]. In particular, the GEP for a symmetric matrix $\mathbf{A} \in \mathbb{R}^{n \times n}$ and a positive definite matrix $\mathbf{B} \in \mathbb{R}^{n \times n}$ is defined as

$$\mathbf{A}\mathbf{v}_i = \lambda_i \mathbf{B}\mathbf{v}_i, \quad i = 1, 2, \ldots, n. \tag{1}$$

Here, $\lambda_i$ represent the generalized eigenvalues, and $\mathbf{v}_i$ denote the corresponding generalized eigenvectors of the matrix pair $(\mathbf{A}, \mathbf{B})$. The eigenvalues are ordered such that $\lambda_1 \geq \lambda_2 \ldots \geq \lambda_n$. Throughout this paper, we focus on the setting of symmetric semi-definite GEPs.

According to the Rayleigh-Ritz theorem [70], the leading generalized eigenvector of $(\mathbf{A}, \mathbf{B})$ is the optimal solution to the following optimization problem:

$$\max_{\mathbf{u} \in \mathbb{R}^n} \mathbf{u}^\top \mathbf{A}\mathbf{u} \quad \text{s.t.} \quad \mathbf{u}^\top \mathbf{B}\mathbf{u} = 1. \tag{2}$$

Or, in an equivalent form with respect to the Rayleigh quotient (up to the transformation of the norm of the optimal solutions):

$$\max_{\mathbf{u} \in \mathbb{R}^n, \mathbf{u} \neq 0} \frac{\mathbf{u}^\top \mathbf{A}\mathbf{u}}{\mathbf{u}^\top \mathbf{B}\mathbf{u}}. \tag{3}$$

---

[*]Corresponding author.

38th Conference on Neural Information Processing Systems (NeurIPS 2024).

Subsequent generalized eigenvectors of $(\mathbf{A}, \mathbf{B})$ can be obtained through methods such as generalizations of Rayleigh quotient iteration [56, 19, 66], subspace style iterations [99, 25, 89], and the QZ method [64, 43, 45, 10].

In many practical applications, the matrices $\mathbf{A}$ and $\mathbf{B}$ may be contaminated by unknown perturbations, and we can only access approximate matrices $\hat{\mathbf{A}}$ and $\hat{\mathbf{B}}$ based on $m$ independent observations. Specifically, we have

$$\hat{\mathbf{A}} = \mathbf{A} + \mathbf{E}, \quad \hat{\mathbf{B}} = \mathbf{B} + \mathbf{F}, \tag{4}$$

where $\mathbf{E}$ and $\mathbf{F}$ are the perturbation matrices. In addition, in recent years, there has been an increasing interest in studying the GEP in the high-dimensional setting where $m \ll n$, with the prominent assumption that the leading generalized eigenvector $\mathbf{v}_1 \in \mathbb{R}^n$ of the uncorrupted matrix pair $(\mathbf{A}, \mathbf{B})$ (*cf.* Eq. (1)) is sparse. This leads to the sparse generalized eigenvalue problem (SGEP), for which we can estimate the underlying signal $\mathbf{v}_1$ based on the approximate matrices $\hat{\mathbf{A}}$ and $\hat{\mathbf{B}}$ by solving the following constrained optimization problem:

$$\max_{\mathbf{u} \in \mathbb{R}^n} \frac{\mathbf{u}^\top \hat{\mathbf{A}} \mathbf{u}}{\mathbf{u}^\top \hat{\mathbf{B}} \mathbf{u}} \quad \text{s.t.} \quad \mathbf{u}^\top \hat{\mathbf{B}} \mathbf{u} \neq 0, \ \|\mathbf{u}\|_0 \leq s, \tag{5}$$

where $\|\mathbf{u}\|_0 = |\{i \ : \ u_i \neq 0\}|$ represents the number of non-zero entries of $\mathbf{u}$ and $s \in \mathbb{N}$ is a parameter for the sparsity level. As mentioned in [84], solving the non-convex optimization problem in Eq. (5) is challenging in the high-dimensional setting because $\hat{\mathbf{B}}$ is singular and not invertible, preventing us from converting the SGEP to the sparse eigenvalue problem and making use of classical algorithms that require taking the inverse of $\hat{\mathbf{B}}$.

Motivated by the remarkable success of deep generative models in a multitude of real-world applications, a new perspective on high-dimensional inverse problems has recently emerged. In this new perspective, the assumption that the underlying signal can be well-modeled by a pre-trained (deep) generative model replaces the common sparsity assumption. Specifically, in the seminal work [4], the authors explore linear inverse problems with generative models and provide sample complexity upper bounds for accurate signal recovery. Furthermore, they presented impressive numerical results on natural image datasets, demonstrating that the utilization of generative models can result in substantial reductions in the required number of measurements compared to sparsity-based methods. Numerous subsequent works have built upon [4], exploring various aspects of inverse problems under generative priors [86, 29, 42, 36, 67, 93, 35, 65].

In this work, we investigate the high-dimensional GEP under generative priors, which we refer to as the generative generalized eigenvalue problem (GGEP). The corresponding optimization problem becomes:

$$\max_{\mathbf{u} \in \mathbb{R}^n} \frac{\mathbf{u}^\top \hat{\mathbf{A}} \mathbf{u}}{\mathbf{u}^\top \hat{\mathbf{B}} \mathbf{u}} \quad \text{s.t.} \quad \mathbf{u}^\top \hat{\mathbf{B}} \mathbf{u} \neq 0, \ \mathbf{u} \in \mathcal{R}(G), \tag{6}$$

where $\mathcal{R}(G)$ denotes the range of a pre-trained generative model $G : \mathbb{R}^k \to \mathbb{R}^n$, with the condition that the input dimension $k$ is much smaller than the output dimension $n$ to enable accurate recovery of the signal using a small number of measurements or observations.

## 1.1 Related Work

This subsection provides a summary of existing works related to Sparse Generalized Eigenvalue Problems (SGEP) and inverse problems with generative priors.

**SGEP:** There have been extensive studies aimed at developing algorithms and providing theoretical guarantees for specific instances of SGEP. For instance, sparse principal component analysis (SPCA) is one of the most widely studied instances of SGEP, with numerous notable solvers in the literature, including the truncated power method [98], Fantope-based convex relaxation method [88], iterative thresholding approach [58], regression-type method [6], and sparse orthogonal iteration pursuit [91], among others [63, 18, 59, 80, 30, 39, 1, 46, 68, 11, 90]. Additionally, significant developments for sparse FDA and sparse CCA, another two popular instances of SGEP, can be found in various works including [24, 95, 69, 57, 7, 15, 27, 79, 60, 13, 14, 44, 94, 20, 83, 96].

There are also many works that propose general approaches for the SGEP, including the projection-free method proposed in [32], the majorization-minimization approaches proposed in [82, 81], the

decomposition algorithm proposed in [97], an inverse-free truncated Rayleigh-Ritz method in [9], the linear programming based sparse estimation method in [75], among others [76, 62, 84, 40, 8, 31].

Among the works related to SGEP, [84] is the most relevant to ours. In [84], the authors proposed a two-stage computational method for SGEP that achieves linear convergence to a point achieving the optimal statistical rate. Their method first computes an initialization vector via convex relaxation, and then refines the initial guess by truncated gradient ascent, with the method for the second stage being referred to as the truncated Rayleigh flow method. Our work generalizes the truncated Rayleigh flow method to the more complex scenario of using generative priors, and we also establish a linear convergence guarantee to an estimated vector with the optimal statistical error.

**Inverse problems with generative priors:** Since the seminal work [4] that investigated linear inverse problems with generative priors, various high-dimensional inverse problems have been studied using generative models. For example, under the generative modeling assumption, one-bit or more general single-index models have been investigated in [92, 38, 50, 52, 73, 41, 12], spiked matrix models have been investigated in [3, 16, 17], and phase retrieval problems have been investigated in [26, 33, 34, 92, 78, 2, 49, 54].

Among the works that study inverse problems using generative models, the recent work [51] is the most relevant to ours. Specifically, in [51], the authors considered generative model-based PCA (GPCA) and proposed a practical projected power (PPower) method. Furthermore, they showed that provided a good initial guess, PPower converges linearly to an estimated vector with the optimal statistical error. However, as pointed out in [84] for the case of sparse priors, due to the singular matrix $\hat{\mathbf{B}}$ of the GEP, the corresponding methods for PCA generally cannot be directly applied to solve GEPs. Therefore, our work, which provides a unified treatment to GEP under generative priors, broadens the scope of [51].

A detailed discussion on the technical novelty of this work compared to [51] and [84] is provided in Section 2.2.

## 1.2 Contributions

The main contributions of this paper can be summarized as follows:

- We provide theoretical guarantees regarding the optimal solutions to the GGEP in Eq. (6). Particularly, we demonstrate that under appropriate conditions, the distance between any optimal solution to Eq. (6) and the underlying signal is roughly of order $O(\sqrt{(k \log L)/m})$, assuming that the generative model $G$ is $L$-Lipschitz continuous with bounded $k$-dimensional inputs. Such a statistical rate is naturally conjectured to be optimal based on the information-theoretic lower bound established in [51] for the simpler GPCA problem.

- We propose an iterative approach to approximately solve the non-convex optimization problem in Eq. (6), which we refer to as the projected Rayleigh flow method (PRFM). We show that PRFM converges linearly to a point achieving the optimal statistical rate under suitable assumptions.

- We have conducted simple proof-of-concept experiments to demonstrate the effectiveness of the proposed projected Rayleigh flow method.

## 1.3 Notation

We use upper and lower case boldface letters to denote matrices and vectors, respectively. We write $[N] = \{1, 2, \cdots, N\}$ for a positive integer $N$, and we use $\mathbf{I}_N$ to denote the identity matrix in $\mathbb{R}^{N \times N}$. A *generative model* is a function $G : \mathcal{D} \to \mathbb{R}^n$, with latent dimension $k$, ambient dimension $n$, and input domain $\mathcal{D} \subseteq \mathbb{R}^k$. We focus on the setting where $k \ll n$. For a set $S \subseteq \mathbb{R}^k$ and a generative model $G : \mathbb{R}^k \to \mathbb{R}^n$, we write $G(S) = \{G(\mathbf{z}) : \mathbf{z} \in S\}$. We define the radius-$r$ ball in $\mathbb{R}^k$ as $B^k(r) := \{\mathbf{z} \in \mathbb{R}^k : \|\mathbf{z}\|_2 \leq r\}$. In addition, we use $\mathcal{R}(G) := G(B^k(r))$ to denote the range of $G$. We use $\|\mathbf{X}\|_{2 \to 2}$ to denote the spectral norm of a matrix $\mathbf{X}$. We use $\lambda_{\min}(\mathbf{X})$ and $\lambda_{\max}(\mathbf{X})$ to denote the minimum and maximum eigenvalues of $\mathbf{X}$ respectively. $\mathcal{S}^{n-1} := \{\mathbf{x} \in \mathbb{R}^n : \|\mathbf{x}\|_2 = 1\}$ represents the unit sphere in $\mathbb{R}^n$. The symbols $C$, $C'$, and $C''$ are absolute constants whose values may differ from line to line. We use standard Landau symbols for asymptotic notations, with the

implied constants in $\Omega(\cdot)$ terms being implicitly assumed to be sufficiently large and the implied constants in $O(\cdot)$ terms being implicitly assumed to be sufficiently small.

## 2 Preliminary

In this section, we formally introduce the problem and overview the main assumptions that we adopt. Additionally, we discuss the technical distinctions of this work in comparison with the two closely related works [84] and [51]. In Appendix A, we show examples of popular statistical and machine-learning problems that can be expressed as GGEP.

### 2.1 Setup and Main Assumptions

Recall that we assume that we have access to a matrix pair $(\hat{\mathbf{A}}, \hat{\mathbf{B}})$ in $\mathbb{R}^{n \times n}$ constructed from $m$ independent observations, satisfying $\hat{\mathbf{A}} = \mathbf{A} + \mathbf{E}, \quad \hat{\mathbf{B}} = \mathbf{B} + \mathbf{F}$ (see Eq. (4)). Here, $\mathbf{E}$ and $\mathbf{F}$ are the unknown perturbation matrices, $(\mathbf{A}, \mathbf{B})$ is the underlying matrix pair with $\mathbf{A}$ symmetric and $\mathbf{B}$ positive definite, and the generalized eigenvalues are $\lambda_i$ with corresponding generalized eigenvectors $\mathbf{v}_i$ respectively (see Eq. (1)). The generalized eigenvalues are ordered such that $\lambda_1 \geq \lambda_2 \geq \ldots \geq \lambda_n$. For brevity, we fix the norms of the $\mathbf{v}_i$ such that $\mathbf{v}_i^\top \mathbf{B} \mathbf{v}_i = 1$ (and $\mathbf{v}_i^\top \mathbf{B} \mathbf{v}_j = 0$ for $i \neq j$). In addition, we assume that we have an $L$-Lipschitz continuous generative model $G : \mathbb{R}^k \to \mathbb{R}^n$ with $k \ll n$ and the radius $r > 0$. Based on this model formulation for GEP and the generative model, we further make the following assumptions.

First, we assume that the generative model $G$ has bounded inputs in $\mathbb{R}^k$ and it is normalized in the sense that its range is contained in the unit sphere of $\mathbb{R}^n$.

**Assumption 2.1.** For the sake of convenience, we assume that the generative model $G$ maps from $B^k(r)$ to $\mathcal{S}^{n-1}$ for some radius $r > 0$.

*Remark* 2.2. Our results can be readily extended to general unnormalized generative models by considering their normalized variants. More specifically, according to [4], for an $\ell$-layer fully connected feed-forward neural network from $\mathbb{R}^k$ to $\mathbb{R}^n$ with popular activation functions such as ReLU and sigmoid, typically, it is $L$-Lipschitz continuous with $L = n^{\Theta(\ell)}$. In addition, as mentioned in [50, 51], for a general unnormalized $L$-Lipschitz continuous generative model $G$, we can consider a corresponding normalized generative model $\tilde{G} : \mathcal{D} \to \mathcal{S}^{n-1}$ with $\mathcal{D} := \{\mathbf{z} \in B_2^k(r) : \|G(\mathbf{z})\|_2 > R_{\min}\}$ for some $R_{\min} > 0$, and $\tilde{G}(\mathbf{z}) = G(\mathbf{z})/\|G(\mathbf{z})\|_2$. Then, we can set $R_{\min}$ to be as small as $1/n^{\Theta(\ell)}$ and $r$ to be as large as $n^{\Theta(\ell)}$ without altering the scaling laws, which renders the dependence on $R_{\min}$ and $r$ very mild.

Next, we impose assumptions on the generalized eigenvalues and the leading generalized eigenvector.

**Assumption 2.3.** We suppose that the largest generalized eigenvalue is strictly greater than the second largest generalized eigenvalue, i.e., $\lambda_1 > \lambda_2$. Furthermore, we assume that the normalized leading generalized eigenvector $\mathbf{v}^* := \mathbf{v}_1/\|\mathbf{v}_1\|_2$ lies in the range of the generative model $G$, meaning $\mathbf{v}^* \in \mathcal{R}(G) \subseteq \mathcal{S}^{n-1}$.

We also make appropriate assumptions on the perturbation matrices $\mathbf{E}$ and $\mathbf{F}$. In particular, following [84, Proposition 2] and [51, Assumption 2], we make the following assumption for the perturbation matrices. Similarly to that mentioned in [84], it is easy to verify that under generative priors, for various statistical models that can be formulated as a GEP such as CCA, FDA, and SDR (discussed in Appendix A), Assumption 2.4 will be satisfied with high probability. Additionally, similarly to the proof for the spiked covariance model in [51, Appendix B.1], it can be readily demonstrated that for the $(\hat{\mathbf{A}}, \hat{\mathbf{B}})$ constructed in Section 4.2, the corresponding perturbation matrices $\mathbf{E}$ and $\mathbf{F}$ satisfy Assumption 2.4 with high probability.

**Assumption 2.4.** Let $S_1, S_2$ be two finite sets in $\mathbb{R}^n$ satisfying $m = \Omega(\log(|S_1| \cdot |S_2|))$. Then, for all $\mathbf{s}_1 \in S_1$ and $\mathbf{s}_2 \in S_2$, we have

$$|\mathbf{s}_1^\top \mathbf{E} \mathbf{s}_2| \leq C\sqrt{\frac{\log(|S_1| \cdot |S_2|)}{m}} \cdot \|\mathbf{s}_1\|_2 \cdot \|\mathbf{s}_2\|_2, \quad |\mathbf{s}_1^\top \mathbf{F} \mathbf{s}_2| \leq C\sqrt{\frac{\log(|S_1| \cdot |S_2|)}{m}} \cdot \|\mathbf{s}_1\|_2 \cdot \|\mathbf{s}_2\|_2, \tag{7}$$

where $C$ is an absolute constant. Moreover, we have $\|\mathbf{E}\|_{2 \to 2} = O(n/m)$ and $\|\mathbf{F}\|_{2 \to 2} = O(n/m)$.

*Remark* 2.5. Assumption 2.4 is closely related to classic assumptions about the Crawford number of the symmetric-definite matrix pair $(\mathbf{A}, \mathbf{B})$ [84, 8]. More specifically, we have

$$\mathrm{cr}(\mathbf{A}, \mathbf{B}) := \min_{\mathbf{u} \in \mathcal{S}^{n-1}} \sqrt{(\mathbf{u}^\top \mathbf{A} \mathbf{u})^2 + (\mathbf{u}^\top \mathbf{B} \mathbf{u})^2} \geq \lambda_{\min}(\mathbf{B}) > 0. \tag{8}$$

Additionally, based on the $L$-Lipschitz continuity of $G$ and [87, Lemma 5.2], for any $\delta > 0$, there exists a $\delta$-net $M'$ of $\mathcal{R}(G)$ such that $\log |M'| \leq k \log \frac{4Lr}{\delta}$. Note that $M' \subseteq \mathcal{R}(G) \subseteq \mathcal{S}^{n-1}$. Then, if

$$\epsilon(M') := \sqrt{\rho(\mathbf{E}, M')^2 + \rho(\mathbf{F}, M')^2}, \tag{9}$$

with

$$\rho(\mathbf{E}, M') := \sup_{\mathbf{s}_1 \in M', \mathbf{s}_2 \in M'} |\mathbf{s}_1^\top \mathbf{E} \mathbf{s}_2|, \quad \rho(\mathbf{F}, M') := \sup_{\mathbf{s}_1 \in M', \mathbf{s}_2 \in M'} |\mathbf{s}_1^\top \mathbf{F} \mathbf{s}_2|, \tag{10}$$

it follows from Assumption 2.4 that

$$\epsilon(M') \leq 2C \sqrt{\frac{k \log \frac{4Lr}{\delta}}{m}}. \tag{11}$$

Therefore, if $m = \Omega\left(k \log \frac{4Lr}{\delta}\right)$ with a sufficiently large implied constant, the conditions that

$$\frac{\epsilon(M')}{\mathrm{cr}(\mathbf{A}, \mathbf{B})} \leq c \quad \text{and} \quad \rho(\mathbf{F}, M') \leq c' \lambda_{\min}(\mathbf{B}) \tag{12}$$

naturally hold, where $c, c'$ are certain positive constants.[2] This leads to an assumption similar to [84, Assumption 1].

## 2.2 Discussion on the Technical Novelty Compared to [51] and [84]

Our analysis builds on techniques from existing works such as [51] and [84], but these techniques are combined and extended in a non-trivial manner; we highlight some examples as follows:

- We present a recovery guarantee regarding the global optimal solutions of GGEP in Theorem 3.1. Such guarantees have not been provided for SGEP in [84]. Although in [51, Theorem 1], the authors established a recovery guarantee regarding the global optimal solutions of GPCA, its proof is significantly simpler than ours and does not involve handling singular $\hat{\mathbf{B}}$ or using the property of projection as in Eq. (78) in our proof.

- The convergence guarantee for Rifle in [84] hinges on the generalized eigenvalue decomposition of $(\hat{\mathbf{A}}_F, \hat{\mathbf{B}}_F)$, where $F$ is a superset of the support of the underlying sparse signal, along with their Lemma 4, which follows directly from [98, Lemma 12] and characterizes the error induced by the truncation step. Additionally, the convergence guarantee of PPower in [51] only involves bounding the term related to the sample covariance matrix $\mathbf{V}$ (as seen in their Eq. (83)). In contrast, in our proof of Theorem 3.3, we make use of the generalized eigenvalue decomposition of $(\mathbf{A}, \mathbf{B})$, and we require the employment of significantly distinct techniques to manage the projection step (and bound a term involving both $\hat{\mathbf{A}}$ and $\hat{\mathbf{B}}$, see our Eq. (111)). This is evidenced in the three auxiliary lemmas we introduced in Appendix B.1 and the appropriate manipulation of the first and second terms on the right-hand side of Eq. (114).

- Unlike in GPCA studied in [51], where the underlying signal can be readily assumed to be a unit vector that is (approximately) within the range of the normalized generative model, in the formulation of the optimization problem for GGEP (refer to Eq. (6) and the corresponding analysis, we need to carefully address the distinct normalization requirements of the generative model and $\hat{\mathbf{B}}$.

---

[2]Note that although we use notations such as $\mathrm{cr}(\mathbf{A}, \mathbf{B})$ and $\lambda_{\min}(\mathbf{B})$, both $\mathbf{A}$ and $\mathbf{B}$ are considered fixed matrices, and for the ease of presentation, we omit the dependence on them for relevant positive constants.

**Algorithm 1** Projected Rayleigh Flow Method (PRFM)

---

**Input**: $\hat{\mathbf{A}}$, $\hat{\mathbf{B}}$, $G$, number of iterations $T$, step size $\eta > 0$, initial vector $\mathbf{u}_0$

**for** $t = 0, 1, \ldots, T-1$ **do**

$$\rho_t = \frac{\mathbf{u}_t^\top \hat{\mathbf{A}} \mathbf{u}_t}{\mathbf{u}_t^\top \hat{\mathbf{B}} \mathbf{u}_t}, \tag{14}$$

$$\mathbf{u}_{t+1} = \mathcal{P}_G \left( \mathbf{u}_t + \eta(\hat{\mathbf{A}} - \rho_t \hat{\mathbf{B}})\mathbf{u}_t \right), \tag{15}$$

**end for**

**Output**: $\mathbf{u}_T$

---

## 3 Main Results

Firstly, we present the following theorem, which pertains to the recovery guarantees in relation to the globally optimal solution of the GGEP in Eq. (6). Similar to [51, Theorem 1], Theorem 3.1 can be easily extended to the case where there is representation error, i.e., the underlying signal $\mathbf{v}^* \notin \mathcal{R}(G)$. Here, we focus on the case where $\mathbf{v}^* \in \mathcal{R}(G)$ (*cf.* Assumption 2.3) to avoid non-essential complications. The proof of Theorem 3.1 is deferred to Appendix B.

**Theorem 3.1.** *Suppose that Assumptions 2.1, 2.3, and 2.4 hold for the GEP and generative model $G$. Let $\hat{\mathbf{u}}$ be a globally optimal solution to Eq.* (6) *for GGEP. Then, for any $\delta \in (0, 1)$ satisfying $\delta = O\big((k \log \frac{4Lr}{\delta})/n\big)$, when $m = \Omega\big(k \log \frac{4Lr}{\delta}\big)$, we have*

$$\min\{\|\hat{\mathbf{u}} - \mathbf{v}^*\|_2, \|\hat{\mathbf{u}} + \mathbf{v}^*\|_2\} \leq C_1 \cdot \sqrt{\frac{k \log \frac{4Lr}{\delta}}{m}}, \tag{13}$$

*where $C_1$ is a positive constant depending on $\mathbf{A}$ and $\mathbf{B}$.*

As noted in [4], an $\ell$-layer neural network generative model is typically $L$-Lipschitz continuous with $L = n^{\Theta(\ell)}$. Then, under the typical scaling of $L = n^{\Theta(\ell)}$, $r = n^{\Theta(\ell)}$, and $\delta = 1/n^{\Theta(\ell)}$, the upper bound in Eq. (13) is of order $O(\sqrt{(k \log L)/m})$, which is naturally conjectured to be optimal based on the algorithm-independent lower bound provided in [51] for the simpler GPCA problem.

Although Theorem 3.1 demonstrates that the estimator for the GGEP in Eq. (6) achieves the optimal statistical rate, in general, the optimization problem is highly non-convex, and obtaining the optimal solution is not feasible. To address this issue, we propose an iterative approach that can be regarded as a generative counterpart of the truncated Rayleigh flow method proposed in [84]. This approach is used to find an estimated vector that approximates a globally optimal solution to Eq. (6). The corresponding algorithm, which we refer to as the projected Rayleigh flow method (PRFM), is presented in Algorithm 1.

In the iterative process of Algorithm 1, the following steps are performed:

- Calculate $\rho_t$ in Eq. (14) to approximate the largest generalized eigenvalue $\lambda_1$. Note that from Lemma B.1 in Appendix B.1, we obtain $\mathbf{u}_t^\top \hat{\mathbf{B}} \mathbf{u}_t > 0$ for $t > 0$ under appropriate conditions.

- In Eq. 15, we perform a gradient ascent operation and a projection operation onto the range of the generative model, where $\mathcal{P}_G(\cdot)$ denotes the projection function. This step is essentially analogous to the corresponding step in [84, Algorithm 1]. However, instead of seeking the support for the sparse signal, our aim is to project onto the range of the generative model. Furthermore, we adopt a simpler choice for the step size $\eta$ in the gradient ascent operation.[3]

*Remark* 3.2. Specifically, for any $\mathbf{u} \in \mathbb{R}^n$, $\mathcal{P}_G(\mathbf{u}) \in \arg\min_{\mathbf{w} \in \mathcal{R}(G)} \|\mathbf{w} - \mathbf{u}\|_2$. We will assume implicitly that the projection step can be performed accurately, as in [77, 71, 51], for the convenience of theoretical analysis. In practice, however, approximate methods might be necessary, such as gradient descent [77] or GAN-based projection methods [74].

---

[3]In [84, Algorithm 1], the step size is $\eta/\rho_t$, which depends on $t$ and is approximately equal to $\eta/\lambda_1$.

We establish the following convergence guarantee for Algorithm 1. The proof of Theorem 3.3 can be found in Appendix C.

**Theorem 3.3.** *Suppose that Assumptions 2.1, 2.3, and 2.4 hold for the GEP and generative model $G$. Let $\gamma_1 = \eta(\lambda_1 - \lambda_2)\lambda_{\min}(\mathbf{B})$ and $\gamma_2 = \eta(\lambda_1 - \lambda_n)\lambda_{\max}(\mathbf{B})$. Suppose that*

$$\gamma_1 + \gamma_2 < 2, \tag{16}$$

*and $\nu_0 := \mathbf{u}_0^\top \mathbf{v}^* > 0$ satisfies the condition that*

$$\frac{b_0 + \sqrt{(1 - c_0)c_0}}{1 - c_0} < 1, \tag{17}$$

*where*

$$b_0 = (2 - (\gamma_1 + \gamma_2)) + \gamma_1(2\kappa(\mathbf{B}) - (1 + \nu_0)) + 3\gamma_2\kappa(\mathbf{B})\sqrt{2(1 - \nu_0)}, \quad c_0 = \frac{\gamma_2 - \gamma_1}{2}. \tag{18}$$

*Then, for any $\delta \in (0, 1)$ satisfying $\delta = O\big((k \log \frac{4Lr}{\delta})/n\big)$, when $m = \Omega\big(k \log \frac{4Lr}{\delta}\big)$, there exists a positive integer $T_0 = O\big(\log\big(\frac{m}{k \log \frac{Lr}{\delta}}\big)\big)$, such that the sequence $\{\|\mathbf{u}_t - \mathbf{v}^*\|_2\}_{t \leq T_0}$ is monotonically decreasing, with the following inequality holds for all $t \leq T_0$:*

$$\|\mathbf{u}_t - \mathbf{v}^*\|_2 \leq \left(\frac{b_0 + \sqrt{(1 - c_0)c_0}}{1 - c_0}\right)^t \cdot \|\mathbf{u}_0 - \mathbf{v}^*\|_2 + C_2\sqrt{\frac{k \log \frac{4Lr}{\delta}}{m}}, \tag{19}$$

*where $C_2$ is a positive constant depending on $\mathbf{A}$, $\mathbf{B}$, and $\eta$. Additionally, we have for all $t \geq T_0$ that*

$$\|\mathbf{u}_t - \mathbf{v}^*\|_2 \leq C_2\sqrt{\frac{k \log \frac{4Lr}{\delta}}{m}}. \tag{20}$$

Theorem 3.3 establishes the conditions under which Algorithm 1 converges linearly to a point that achieves the statistical rate of order $O(\sqrt{(k \log L)/m})$.

*Remark 3.4.* Both inequalities in Eqs. (16) and (17) can be satisfied under appropriate conditions. More specifically, first, under suitable conditions on the underlying matrix pair $(\mathbf{A}, \mathbf{B})$ and the step size $\eta$, the condition in (16) can hold. Moreover, using of the inequality that $2\sqrt{(1 - c)c} \leq 1$ for any $c \in [0, 1)$, we obtain that when

$$\frac{2b_0 + 1}{2(1 - c_0)} < 1, \tag{21}$$

or equivalently,

$$\gamma_2 + 3\gamma_1 - 2\gamma_1(2\kappa(\mathbf{B}) - (1 + \nu_0)) - 6\gamma_2\kappa(\mathbf{B})\sqrt{2(1 - \nu_0)} > 3, \tag{22}$$

the condition in Eq. (17) holds. Then, for example, when $\kappa(\mathbf{B}) := \lambda_{\max}(\mathbf{B})/\lambda_{\min}(\mathbf{B})$ is close to 1, and $\nu_0$ is close to 1, (22) can be approximately simplified as

$$\gamma_2 + 3\gamma_1 > 3. \tag{23}$$

Note that both the conditions $\gamma_1 + \gamma_2 < 2$ (in Eq. (16)) and $\gamma_2 + 3\gamma_1 > 3$ can be satisfied for appropriate $\gamma_1$ and $\gamma_2$ (under suitable conditions for $\mathbf{A}$, $\mathbf{B}$, and $\eta$), say $\gamma_1 = \frac{2}{3}$ and $\gamma_2 = 1.1$.

*Remark 3.5.* In certain practical scenarios, we may assume that the data only contains non-negative vectors, for example, in the case of image datasets. Additionally, during pre-training, we can set the activation function of the final layer of the neural network generative model to be a non-negative function, such as ReLU or sigmoid, further restricting the range of the generative model to the non-negative orthant. Therefore, the assumption that $\nu_0 := \mathbf{u}_0^\top \mathbf{v}^* > 0$ is mild (in experiments, we simply set the initial vector $\mathbf{u}_0$ to be $\mathbf{u}_0 = [1, 1, \ldots, 1]^\top/\sqrt{n} \in \mathbb{R}^n$), and similar assumptions have been made in prior works such as [51]. As a result, we provide an upper bound on $\|\mathbf{u}_t - \mathbf{v}^*\|_2$ instead of on $\min\{\|\mathbf{u}_t - \mathbf{v}^*\|_2, \|\mathbf{u}_t + \mathbf{v}^*\|_2\}$.

# 4 Experiments

In this section, we conduct proof-of-concept numerical experiments on the MNIST dataset [47] to showcase the effectiveness of the proposed Algorithm 1. Additional results for MNIST and CelebA [55] are provided in Appendices D and E.[4] We note that these experiments are intended as a basic proof of concept rather than an exhaustive study, as our contributions are primarily theoretical in nature.

## 4.1 Experiment Setup

The MNIST dataset consists of $60,000$ images of handwritten digits, each measuring $28 \times 28$ pixels, resulting in an ambient dimension of $n = 784$. We choose a pre-trained variational autoencoder (VAE) model as the generative model $G$ for the MNIST dataset, with a latent dimension of $k = 20$. Both the encoder and decoder of the VAE are fully connected neural networks with two hidden layers, having an architecture of $20 - 500 - 500 - 784$. The VAE is trained using the Adam optimizer with a mini-batch size of 100 and a learning rate of 0.001 on the original MNIST training set. To approximately perform the projection step $\mathcal{P}_G(\cdot)$, we use a gradient descent method with the Adam optimizer, with a step size of $100$ and a learning rate of $0.1$. This approximation method has been used in several previous works, including [77, 71, 50, 51]. The reconstruction task is evaluated on a random subset of 10 images drawn from the testing set of the MNIST dataset, which is unseen by the pre-trained generative model.

We compare our Algorithm 1 (denoted as PRFM) with the projected power method (denoted as PPower) proposed in [51] and the Rifle method (e.g., denoted as Rifle20 when the cardinality parameter is set to 20) proposed in [84].

To evaluate the performance of different algorithms, we employ the Cosine Similarity metric, which is calculated as $CosSim(\mathbf{v}^*, \tilde{\mathbf{u}}) = \tilde{\mathbf{u}}^\top \mathbf{v}^*$. Here, $\mathbf{v}^*$ is the target signal that is contained in the unit sphere, and $\tilde{\mathbf{u}}$ denotes the normalized output vector of each algorithm. To mitigate the effect of local minima, we perform 10 random restarts and select the best result from these restarts. The average Cosine Similarity is calculated over the 10 test images and the 10 restarts. All experiments are carried out using Python 3.10.6 and PyTorch 2.0.0, with an NVIDIA RTX 3060 Laptop 6GB GPU.

## 4.2 Results for GGEPs

Firstly, we adhere to the experimental setup employed in [8], where the underlying matrices $\mathbf{A}$ and $\mathbf{B}$ are set to be $\mathbf{A} = 4\mathbf{v}^*(\mathbf{v}^*)^\top + \mathbf{I}_n$ and $\mathbf{B} = \mathbf{I}_n$ respectively. We sample $m$ data points following $\mathcal{N}(\mathbf{0}, \mathbf{A})$, and another $m$ data points following $\mathcal{N}(\mathbf{0}, \mathbf{B})$. The approximate matrices $\hat{\mathbf{A}}$ and $\hat{\mathbf{B}}$ are constructed based on the sample covariances correspondingly. Specifically, we set

$$\hat{\mathbf{A}} = \frac{1}{m} \sum_{i=1}^m (2\gamma_i \mathbf{v}^* + \mathbf{z}_i)(2\gamma_i \mathbf{v}^* + \mathbf{z}_i)^\top, \quad \hat{\mathbf{B}} = \frac{1}{m} \sum_{i=1}^m \mathbf{w}_i \mathbf{w}_i^\top. \tag{24}$$

Here, $\gamma_i$ are independently and identically distributed (i.i.d.) realizations of the standard normal distribution $\mathcal{N}(0, 1)$, $\mathbf{z}_i$ are i.i.d. realizations of $\mathcal{N}(\mathbf{0}, \mathbf{I}_n)$, and $\mathbf{w}_i$ are also i.i.d. realizations of $\mathcal{N}(\mathbf{0}, \mathbf{I}_n)$. $\alpha_i$, $\mathbf{z}_i$, and $\mathbf{w}_i$ are independently generated. The leading generalized eigenvector $\mathbf{v}^*$ is set to be the normalized version of the test image vector. Note that in this case, the generalized eigenvalues are $\lambda_1 = 5$ and $\lambda_2 = \ldots = \lambda_n = 1$. To ensure that the conditions in Eqs. (16) and (23) are both satisfied, we set the step size $\eta$ for PRFM to be $\eta = \frac{7}{32}$. The step size $\eta'$ for Rifle is set to $35/32$ such that $\eta'/\rho_{t-1} \approx 7/32 = \eta$. For all the methods, the initialization vector $\mathbf{u}_0$ for is set to be the normalized vector of all ones, namely $\mathbf{u}_0 = [1, 1, \ldots, 1]^\top / \sqrt{n} \in \mathbb{R}^n$, which naturally guarantees that $\nu_0 = \mathbf{u}_0^\top \mathbf{v}^* > 0$ since the image vectors in the MNIST dataset contain only non-negative entries. For PPower, the input matrix set to be $\hat{\mathbf{A}}$ (ignoring the fact that $\hat{\mathbf{B}}$ is not exactly the identity matrix; see [51, Algorithm 1]). We vary the number of measurements $m$ in $\{100, 150, 200, 250, 300, 350\}$.

Figure 1(a) show the reconstructed images with different numbers of measurements. Additionally, Figure 2(a) presents the quantitative comparison results based on the Cosine Similarity metric. From

---

[4]For the CelebA dataset, which is publicly accessible and widely used (containing face images of celebrities), we point out the potential ethical problems in Appendix E.

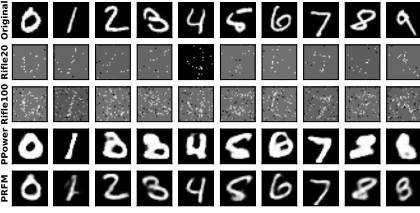 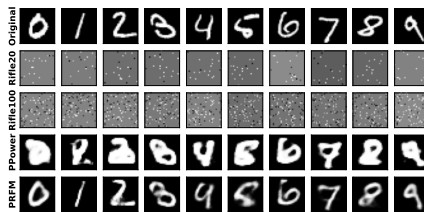

(a) Eq. (24) with $m = 150$          (b) Eq. (25) with $m = 300$

Figure 1: Reconstructed images of the MNIST dataset for $(\hat{\mathbf{A}}, \hat{\mathbf{B}})$ generated from Eqs. (24) and (25).

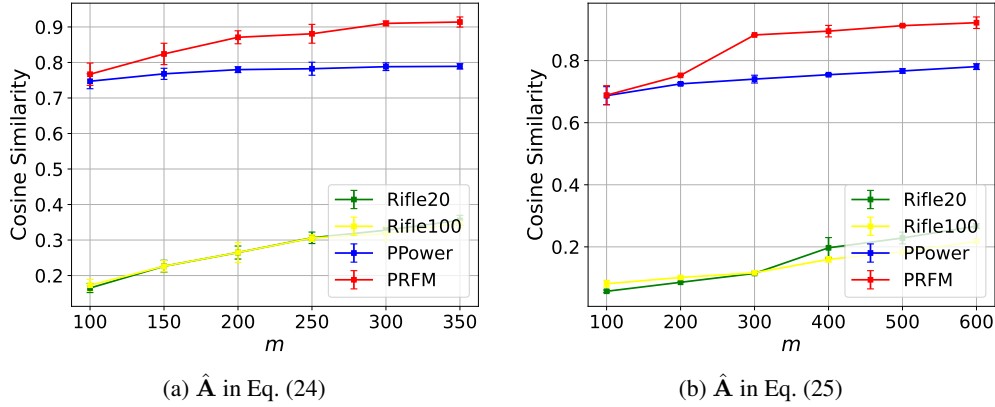

(a) $\hat{\mathbf{A}}$ in Eq. (24)          (b) $\hat{\mathbf{A}}$ in Eq. (25)

Figure 2: Quantitative results of the performance of the methods on MNIST.

these figures, we can see that when the number of measurements $m$ is relatively small compared to the ambient dimension $n$, sparsity-based methods `Rifle20` and `Rifle100` lead to poor reconstructions, and `PPower` and `PRFM` can achieve reasonably good reconstructions. Moreover, `PRFM` generally yields significantly better results than `PPower`. This is not surprising as `PPower` is not compatible with the case where $\hat{\mathbf{B}}$ is not the identity matrix.

Next, we also investigate the case where

$$\hat{\mathbf{A}} = \frac{1}{m} \sum_{i=1}^{m} y_i \mathbf{g}_i \mathbf{g}_i^{\top}, \quad \hat{\mathbf{B}} = \frac{1}{m} \sum_{i=1}^{m} \mathbf{w}_i \mathbf{w}_i^{\top}, \tag{25}$$

where $\mathbf{g}_i \in \mathbb{R}^n$ are independent standard Gaussian vectors and $y_i = (\mathbf{g}_i^{\top} \mathbf{v}^*)^2$. The generation of $\hat{\mathbf{A}}$ corresponds to the phase retrieval model. Other experimental settings remain the same as those in the previous case where $\hat{\mathbf{A}}$ and $\hat{\mathbf{B}}$ are generated from Eq. (24). The corresponding numerical results are presented in Figures 1(b) and 2(b). From these figures, we can see that `PRFM` also leads to best reconstructions in this case.

## 5 Conclusion and Future Work

GEP encompasses numerous significant eigenvalue problems, motivating this paper's examination of GEP using generative priors, referred to as Generative GEP and GGEP for brevity. Specifically, we assume that the desired signal lies within the range of a specific pre-trained Lipschitz generative model. Unlike prior works that typically addressed high-dimensional settings through sparsity, the generative prior enables the accurate characterization of more intricate signals. We have demonstrated that the exact solver of GGEP attains the optimal statistical rate. Furthermore, we have devised a computational method that converges linearly to a point achieving the optimal rate under appropriate assumptions. Experimental results are provided to demonstrate the efficacy of our method.

In the current work, for the sake of theoretical analysis, we adhere to the assumption made in prior works like [77, 71, 51] that the projection onto the range of the generative model can be executed accurately. However, in experiments, this projection step can only be approximated, consuming a substantial portion of the running time of our proposed method. Developing highly efficient projection methods with theoretical guarantees is of both theoretical and practical interest. Another intriguing area for future research is to provide guarantees for estimating multiple generalized eigenvectors (or the subspace they span) under generative modeling assumptions.

## Acknowledgments and Disclosure of Funding

Z. Liu and W. Li were supported by the National Natural Science Foundation of China (No. 62176047), the Sichuan Science and Technology Program (No. 2022YFS0600), Sichuan Natural Science Foundation (No. 2024NSFTD0041), and the Fundamental Research Funds for the Central Universities Grant (No. ZYGX2021YGLH208). J. Chen was supported by a Hong Kong PhD Fellowship from the Hong Kong Research Grants Council (RGC).

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

# A  Instances of GGEPs

GEP encompasses a broad range of eigenvalue problems, and in the following, we specifically discuss three noteworthy examples under generative priors.

- **Generative Principle Component Analysis (GPCA)**. Given a data matrix $\hat{\boldsymbol{\Sigma}}$ that typically represents the covariance matrix of $m$ observed data points with $n$ features, the goal of PCA is to find a projection that maximizes variance. Under the generative prior, the generative PCA (GPCA) problem can be formulated as follows (assuming that the range of the generative model $G$ is a subset of the unit sphere for simplicity)

$$\max_{\mathbf{u} \in \mathbb{R}^n} \ \mathbf{u}^\top \hat{\boldsymbol{\Sigma}} \mathbf{u} \quad \text{s.t.} \quad \mathbf{u} \in \mathcal{R}(G). \tag{26}$$

  This problem can be derived from the optimization problem for GGEP in Eq. (6) with $\hat{\mathbf{A}} = \hat{\boldsymbol{\Sigma}}$ and $\hat{\mathbf{B}} = \mathbf{I}_n$.

- **Generative Fisher's Discriminant Analysis (GFDA)**. Given $m$ observations with $n$ features belonging to $K$ different classes, Fisher's discriminant analysis (FDA) aims to find a low-dimensional projection that maps the observations to a lower-dimensional space, where the between-class variance $\boldsymbol{\Sigma}_b$ is large while the within-class variance $\boldsymbol{\Sigma}_w$ is relatively small. Let $(\hat{\boldsymbol{\Sigma}}_b, \hat{\boldsymbol{\Sigma}}_w)$ be the empirical version of $(\boldsymbol{\Sigma}_b, \boldsymbol{\Sigma}_w)$ constructed from the observations. Under generative priors, the goal of generative FDA (GFDA) is to solve the following optimization problem:

$$\max_{\mathbf{u} \in \mathbb{R}^n} \ \frac{\mathbf{u}^\top \hat{\boldsymbol{\Sigma}}_b \mathbf{u}}{\mathbf{u}^\top \hat{\boldsymbol{\Sigma}}_w \mathbf{u}} \quad \text{s.t.} \quad \mathbf{u}^\top \hat{\boldsymbol{\Sigma}}_w \mathbf{u} \neq 0, \ \mathbf{u} \in \mathcal{R}(G), \tag{27}$$

  which can be derived from the optimization problem for GGEP in Eq. (6) with $\hat{\mathbf{A}} = \hat{\boldsymbol{\Sigma}}_b$ and $\hat{\mathbf{B}} = \hat{\boldsymbol{\Sigma}}_w$.

- **Generative Canonical Correlation Analysis (GCCA)**. Given random vectors $\mathbf{x}, \mathbf{y} \in \mathbb{R}^p$ and their realizations, let $\boldsymbol{\Sigma}_{xx}$ and $\boldsymbol{\Sigma}_{yy}$ be the corresponding covariance matrices, $\boldsymbol{\Sigma}_{xy}$ be the cross-covariance matrix between $\mathbf{x}$ and $\mathbf{y}$, and $\tilde{\boldsymbol{\Sigma}}_{xx}, \tilde{\boldsymbol{\Sigma}}_{yy}, \tilde{\boldsymbol{\Sigma}}_{xy}$ be the empirical version of $\boldsymbol{\Sigma}_{xx}, \boldsymbol{\Sigma}_{yy}, \boldsymbol{\Sigma}_{xy}$, respectively. Then, canonical correlation analysis aims to solve the following optimization problem:

$$\max_{\mathbf{u}_x, \mathbf{u}_y} \ \mathbf{u}_x^\top \tilde{\boldsymbol{\Sigma}}_{xy} \mathbf{u}_y$$
$$\text{s.t.} \quad \mathbf{u}_x^\top \tilde{\boldsymbol{\Sigma}}_{xx} \mathbf{u}_x = 1, \quad \mathbf{u}_y^\top \tilde{\boldsymbol{\Sigma}}_{yy} \mathbf{u}_y = 1. \tag{28}$$

  The optimization problem in Eq. (28) can be equivalently formulated as

$$\max_{\mathbf{u} \in \mathbb{R}^n} \ \frac{\mathbf{u}^\top \hat{\mathbf{A}} \mathbf{u}}{\mathbf{u}^\top \hat{\mathbf{B}} \mathbf{u}} \quad \text{s.t.} \quad \mathbf{u}^\top \hat{\mathbf{B}} \mathbf{u} \neq 0, \tag{29}$$

  where

$$\hat{\mathbf{A}} := \begin{bmatrix} 0 & \tilde{\boldsymbol{\Sigma}}_{xy} \\ \tilde{\boldsymbol{\Sigma}}_{xy}^\top & 0 \end{bmatrix}, \ \hat{\mathbf{B}} := \begin{bmatrix} \tilde{\boldsymbol{\Sigma}}_{xx} & 0 \\ 0 & \tilde{\boldsymbol{\Sigma}}_{yy} \end{bmatrix}, \tag{30}$$

  and $\mathbf{u} = \begin{bmatrix} \mathbf{u}_x \\ \mathbf{u}_y \end{bmatrix}$. Imposing the generative prior on $\mathbf{u}$ and adding the corresponding constraint into Eq. (29) (leading to the optimization problem for GGEP in Eq. (6)), we derive generative CCA (GCCA).

# B  Proof of Theorem 3.1 (Guarantees for Globally Optimal Solutions)

## B.1  Auxiliary Lemmas for the Proof of Theorem 3.1

We present the following useful lemma, which shows that for any globally optimal solution $\hat{\mathbf{u}}$ to GGEP, we have $\hat{\mathbf{u}}^\top \hat{\mathbf{B}} \hat{\mathbf{u}} > 0$ under appropriate conditions.

**Lemma B.1.** *Suppose that Assumptions 2.1, 2.3, and 2.4 hold for the GEP and generative model G. Let $\hat{\mathbf{u}}$ be a globally optimal solution to Eq.* (6) *for GGEP. Then, for any $\delta \in (0,1)$ satisfying $\delta = O(m/n)$ and $m = \Omega\left(k \log \frac{4Lr}{\delta}\right)$, we have*

$$\hat{\mathbf{u}}^\top \hat{\mathbf{B}} \hat{\mathbf{u}} > 0. \tag{31}$$

*Proof.* For any $\mathbf{u} \in \mathcal{S}^{n-1}$, we have $\mathbf{u}^\top \mathbf{B} \mathbf{u} \geq \lambda_{\min}(\mathbf{B})$, where $\lambda_{\min}(\mathbf{B}) > 0$ denotes the smallest eigenvalue of the positive definite matrix $\mathbf{B}$. Additionally, let $M$ be a $(\delta/L)$-net of $B^k(r)$. From [87, Lemma 5.2], we know that there exists such a net with

$$\log |M| \leq k \log \frac{4Lr}{\delta}. \tag{32}$$

Since $G$ is $L$-Lipschitz continuous, we have that $M' := G(M)$ is a $\delta$-net of $\mathcal{R}(G) = G(B_2^k(r))$. Note that $M' \subseteq \mathcal{R}(G) \subseteq \mathcal{S}^{n-1}$. Since $\hat{\mathbf{u}} \in \mathcal{R}(G)$, there exists a $\mathbf{w} \in M'$ such that $\|\hat{\mathbf{u}} - \mathbf{w}\|_2 \leq \delta$. Then,

$$\hat{\mathbf{u}}^\top \hat{\mathbf{B}} \hat{\mathbf{u}} = (\hat{\mathbf{u}} - \mathbf{w} + \mathbf{w})^\top \hat{\mathbf{B}} (\hat{\mathbf{u}} - \mathbf{w} + \mathbf{w}) \tag{33}$$

$$= \mathbf{w}^\top \hat{\mathbf{B}} \mathbf{w} + (\hat{\mathbf{u}} - \mathbf{w})^\top \hat{\mathbf{B}} \mathbf{w} + \mathbf{w}^\top \hat{\mathbf{B}} (\hat{\mathbf{u}} - \mathbf{w}) + (\hat{\mathbf{u}} - \mathbf{w})^\top \hat{\mathbf{B}} (\hat{\mathbf{u}} - \mathbf{w}) \tag{34}$$

$$= \mathbf{w}^\top \mathbf{B} \mathbf{w} + \mathbf{w}^\top \mathbf{F} \mathbf{w} + (\hat{\mathbf{u}} - \mathbf{w})^\top \hat{\mathbf{B}} \mathbf{w} + \mathbf{w}^\top \hat{\mathbf{B}} (\hat{\mathbf{u}} - \mathbf{w}) + (\hat{\mathbf{u}} - \mathbf{w})^\top \hat{\mathbf{B}} (\hat{\mathbf{u}} - \mathbf{w}). \tag{35}$$

Since $\mathbf{w} \in M'$ is a unit vector, we have $\mathbf{w}^\top \mathbf{B} \mathbf{w} \geq \lambda_{\min}(\mathbf{B})$. Using Assumption 2.4 with $S_1$ and $S_2$ both set to $M'$, we obtain $|\mathbf{w}^\top \mathbf{F} \mathbf{w}| \leq C\sqrt{\frac{k \log \frac{4Lr}{\delta}}{m}}$. Additionally, from Assumption 2.4, we have the upper bound $\|\hat{\mathbf{B}}\|_{2 \to 2} = \|\mathbf{B} + \mathbf{F}\|_{2 \to 2} \leq \lambda_{\max}(\mathbf{B}) + C'\frac{n}{m}$.

Therefore, we obtain that when $m = \Omega\left(k \log \frac{4Lr}{\delta}\right)$ (with a sufficiently large implied positive constant) and $\delta = O(\frac{m}{n})$ (with a sufficiently small implied positive constant), it follows from (35) that[5]

$$\hat{\mathbf{u}}^\top \hat{\mathbf{B}} \hat{\mathbf{u}} \geq \lambda_{\min}(\mathbf{B}) - C\sqrt{\frac{k \log \frac{4Lr}{\delta}}{m}} - \delta(2 + \delta)\left(\lambda_{\max}(\mathbf{B}) + C'\frac{n}{m}\right) > 0. \tag{36}$$

$\square$

### B.2   Complete Proof of Theorem 3.1

First, since $\dim(\text{span}\{\mathbf{v}_1, \mathbf{v}_2, \ldots, \mathbf{v}_n\}) = n$, we know that there exist coefficients $g_i$ such that the unit vector $\hat{\mathbf{u}}$ can be written as

$$\hat{\mathbf{u}} = \sum_{i=1}^n g_i \mathbf{v}_i. \tag{37}$$

Let $d = 1/\|\mathbf{v}_1\|_2$. We have $\mathbf{v}^* = d\mathbf{v}_1 = \frac{\mathbf{v}_1}{\|\mathbf{v}_1\|_2} \in \mathcal{R}(G)$. Additionally, we have the following:

$$\frac{(\mathbf{v}^*)^\top \mathbf{A} \mathbf{v}^*}{(\mathbf{v}^*)^\top \mathbf{B} \mathbf{v}^*} - \frac{\hat{\mathbf{u}}^\top \mathbf{A} \hat{\mathbf{u}}}{\hat{\mathbf{u}}^\top \mathbf{B} \hat{\mathbf{u}}} = \lambda_1 - \frac{\sum_{i=1}^n \lambda_i g_i^2}{\sum_{i=1}^n g_i^2} \tag{38}$$

$$= \frac{\sum_{i=2}^n (\lambda_1 - \lambda_i) g_i^2}{\sum_{i=1}^n g_i^2} \tag{39}$$

$$\geq \frac{(\lambda_1 - \lambda_2) \sum_{i=2}^n g_i^2}{\sum_{i=1}^n g_i^2}. \tag{40}$$

Since $\mathbf{v}_i^\top \mathbf{B} \mathbf{v}_i = 1$ and for $i \neq j$, $\mathbf{v}_i^\top \mathbf{B} \mathbf{v}_j = 0$, if letting $\tilde{\mathbf{v}}_i = \mathbf{B}^{1/2} \mathbf{v}_i$, we observe that $\tilde{\mathbf{v}}_1, \ldots, \tilde{\mathbf{v}}_n$ form an orthonormal basis of $\mathbb{R}^n$. Then, if setting $\tilde{\mathbf{u}} = \mathbf{B}^{1/2} \hat{\mathbf{u}} = \sum_{i=1}^n g_i \tilde{\mathbf{v}}_i$, we have

$$\sum_{i=1}^n g_i^2 = \|\tilde{\mathbf{u}}\|_2^2 = \left\|\mathbf{B}^{1/2}\hat{\mathbf{u}}\right\|_2^2 \leq \lambda_{\max}(\mathbf{B}). \tag{41}$$

---

[5]Note that as mentioned in the footnote at the end of Remark 2.5, $\mathbf{A}$ and $\mathbf{B}$ are considered fixed matrices, and for brevity, we omit the dependence on them for the relevant positive constants.

Combining Eqs. (40) and (41), we obtain

$$\frac{(\mathbf{v}^*)^\top \mathbf{A}\mathbf{v}^*}{(\mathbf{v}^*)^\top \mathbf{B}\mathbf{v}^*} - \frac{\hat{\mathbf{u}}^\top \mathbf{A}\hat{\mathbf{u}}}{\hat{\mathbf{u}}^\top \mathbf{B}\hat{\mathbf{u}}} \geq \frac{(\lambda_1 - \lambda_2) \sum_{i=2}^n g_i^2}{\lambda_{\max}(\mathbf{B})}. \tag{42}$$

In addition, we have

$$\frac{(\mathbf{v}^*)^\top \mathbf{A}\mathbf{v}^*}{(\mathbf{v}^*)^\top \mathbf{B}\mathbf{v}^*} - \frac{\hat{\mathbf{u}}^\top \mathbf{A}\hat{\mathbf{u}}}{\hat{\mathbf{u}}^\top \mathbf{B}\hat{\mathbf{u}}} = \frac{((\mathbf{v}^*)^\top \mathbf{A}\mathbf{v}^*)(\hat{\mathbf{u}}^\top \mathbf{B}\hat{\mathbf{u}}) - (\hat{\mathbf{u}}^\top \mathbf{A}\hat{\mathbf{u}})((\mathbf{v}^*)^\top \mathbf{B}\mathbf{v}^*)}{((\mathbf{v}^*)^\top \mathbf{B}\mathbf{v}^*)(\hat{\mathbf{u}}^\top \mathbf{B}\hat{\mathbf{u}})}, \tag{43}$$

with $(\mathbf{v}^*)^\top \mathbf{B}\mathbf{v}^* = d^2$, $\hat{\mathbf{u}}^\top \mathbf{B}\hat{\mathbf{u}} \geq \lambda_{\min}(\mathbf{B})$, and the numerator $((\mathbf{v}^*)^\top \mathbf{A}\mathbf{v}^*)(\hat{\mathbf{u}}^\top \mathbf{B}\hat{\mathbf{u}}) - (\hat{\mathbf{u}}^\top \mathbf{A}\hat{\mathbf{u}})((\mathbf{v}^*)^\top \mathbf{B}\mathbf{v}^*)$ satisfies

$$\begin{aligned}
&((\mathbf{v}^*)^\top \mathbf{A}\mathbf{v}^*)(\hat{\mathbf{u}}^\top \mathbf{B}\hat{\mathbf{u}}) - (\hat{\mathbf{u}}^\top \mathbf{A}\hat{\mathbf{u}})((\mathbf{v}^*)^\top \mathbf{B}\mathbf{v}^*) \\
&= ((\mathbf{v}^*)^\top (\hat{\mathbf{A}} - \mathbf{E})\mathbf{v}^*)(\hat{\mathbf{u}}^\top (\hat{\mathbf{B}} - \mathbf{F})\hat{\mathbf{u}}) - (\hat{\mathbf{u}}^\top (\hat{\mathbf{A}} - \mathbf{E})\hat{\mathbf{u}})((\mathbf{v}^*)^\top (\hat{\mathbf{B}} - \mathbf{F})\mathbf{v}^*) \qquad (44) \\
&\leq ((\mathbf{v}^*)^\top \mathbf{E}\mathbf{v}^*)(\hat{\mathbf{u}}^\top \mathbf{F}\hat{\mathbf{u}}) - (\hat{\mathbf{u}}^\top \mathbf{E}\hat{\mathbf{u}})((\mathbf{v}^*)^\top \mathbf{F}\mathbf{v}^*) \\
&\quad - ((\mathbf{v}^*)^\top \hat{\mathbf{A}}\mathbf{v}^*)(\hat{\mathbf{u}}^\top \mathbf{F}\hat{\mathbf{u}}) + (\hat{\mathbf{u}}^\top \hat{\mathbf{A}}\hat{\mathbf{u}})((\mathbf{v}^*)^\top \mathbf{F}\mathbf{v}^*) \\
&\quad - ((\mathbf{v}^*)^\top \mathbf{E}\mathbf{v}^*)(\hat{\mathbf{u}}^\top \hat{\mathbf{B}}\hat{\mathbf{u}}) + (\hat{\mathbf{u}}^\top \mathbf{E}\hat{\mathbf{u}})((\mathbf{v}^*)^\top \hat{\mathbf{B}}\mathbf{v}^*), \qquad (45)
\end{aligned}$$

where Eq. (45) follows from the conditions that $\hat{\mathbf{u}}$ is an optimal solution to Eq. (6) and $\mathbf{v}^* \in \mathcal{R}(G)$ (note that it follows from Lemma B.1 that $\hat{\mathbf{u}}^\top \hat{\mathbf{B}}\hat{\mathbf{u}} > 0$ and $(\mathbf{v}^*)^\top \hat{\mathbf{B}}\mathbf{v}^* > 0$). Let $M$ be a $(\delta/L)$-net of $B^k(r)$. From [87, Lemma 5.2], we know that there exists such a net with

$$\log |M| \leq k \log \frac{4Lr}{\delta}. \tag{46}$$

Note that since $G$ is $L$-Lipschitz continuous, we have that $G(M)$ is a $\delta$-net of $\mathcal{R}(G) = G(B_2^k(r))$. Then, we can write $\hat{\mathbf{u}}$ as

$$\hat{\mathbf{u}} = (\hat{\mathbf{u}} - \bar{\mathbf{u}}) + \bar{\mathbf{u}}, \tag{47}$$

where $\bar{\mathbf{u}} \in G(M)$ satisfies $\|\hat{\mathbf{u}} - \bar{\mathbf{u}}\|_2 \leq \delta$. For the first term in the right-hand side of Eq. (45), we have[6]

$$\begin{aligned}
&((\mathbf{v}^*)^\top \mathbf{E}\mathbf{v}^*)(\hat{\mathbf{u}}^\top \mathbf{F}\hat{\mathbf{u}}) - (\hat{\mathbf{u}}^\top \mathbf{E}\hat{\mathbf{u}})((\mathbf{v}^*)^\top \mathbf{F}\mathbf{v}^*) \\
&= ((\mathbf{v}^*)^\top \mathbf{E}\mathbf{v}^*)(\hat{\mathbf{u}}^\top \mathbf{F}\hat{\mathbf{u}}) - ((\mathbf{v}^*)^\top \mathbf{E}\mathbf{v}^*)((\mathbf{v}^*)^\top \mathbf{F}\mathbf{v}^*) \\
&\quad + ((\mathbf{v}^*)^\top \mathbf{E}\mathbf{v}^*)((\mathbf{v}^*)^\top \mathbf{F}\mathbf{v}^*) - (\hat{\mathbf{u}}^\top \mathbf{E}\hat{\mathbf{u}})((\mathbf{v}^*)^\top \mathbf{F}\mathbf{v}^*) \qquad (48) \\
&= ((\mathbf{v}^*)^\top \mathbf{E}\mathbf{v}^*) \cdot (\hat{\mathbf{u}} - \mathbf{v}^*)^\top \mathbf{F}(\hat{\mathbf{u}} + \mathbf{v}^*) + ((\mathbf{v}^*)^\top \mathbf{F}\mathbf{v}^*) \cdot (\hat{\mathbf{u}} - \mathbf{v}^*)^\top \mathbf{E}(\hat{\mathbf{u}} + \mathbf{v}^*). \qquad (49)
\end{aligned}$$

From Assumption 2.4 and Eq. (46), we obtain

$$\left| (\mathbf{v}^*)^\top \mathbf{E}\mathbf{v}^* \right| \leq C \sqrt{\frac{k \log \frac{4Lr}{\delta}}{m}}. \tag{50}$$

Additionally, we have

$$\left| (\hat{\mathbf{u}} - \mathbf{v}^*)^\top \mathbf{F}(\hat{\mathbf{u}} + \mathbf{v}^*) \right| = \left| (\hat{\mathbf{u}} - \bar{\mathbf{u}} + \bar{\mathbf{u}} - \mathbf{v}^*)^\top \mathbf{F}(\hat{\mathbf{u}} - \bar{\mathbf{u}} + \bar{\mathbf{u}} + \mathbf{v}^*) \right| \tag{51}$$

$$\leq \left| (\hat{\mathbf{u}} - \bar{\mathbf{u}})^\top \mathbf{F}(\hat{\mathbf{u}} - \bar{\mathbf{u}}) \right| + \left| (\hat{\mathbf{u}} - \bar{\mathbf{u}})^\top \mathbf{F}(\bar{\mathbf{u}} + \mathbf{v}^*) \right| + \left| (\bar{\mathbf{u}} - \mathbf{v}^*)^\top \mathbf{F}(\hat{\mathbf{u}} - \bar{\mathbf{u}}) \right|$$

$$+ \left| (\bar{\mathbf{u}} - \mathbf{v}^*)^\top \mathbf{F}(\bar{\mathbf{u}} + \mathbf{v}^*) \right| \tag{52}$$

$$\leq \frac{C'n\delta}{m} + \left| (\bar{\mathbf{u}} - \mathbf{v}^*)^\top \mathbf{F}(\bar{\mathbf{u}} + \mathbf{v}^*) \right| \tag{53}$$

$$\leq \frac{C'n\delta}{m} + C \sqrt{\frac{k \log \frac{4Lr}{\delta}}{m}} \cdot \|\bar{\mathbf{u}} - \mathbf{v}^*\|_2 \cdot \|\bar{\mathbf{u}} + \mathbf{v}^*\|_2, \tag{54}$$

---

[6]For brevity, throughout the following, we assume that both $\mathbf{E}$ and $\mathbf{F}$ are symmetric, and we make use of the equality that for any $\mathbf{s}_1, \mathbf{s}_2 \in \mathbb{R}^n$ and any symmetric $\mathbf{G} \in \mathbb{R}^{n\times n}$, $\mathbf{s}_1^T \mathbf{G}\mathbf{s}_1 - \mathbf{s}_2^T \mathbf{G}\mathbf{s}_2 = (\mathbf{s}_1 + \mathbf{s}_2)^T \mathbf{G}(\mathbf{s}_1 - \mathbf{s}_2)$. For the case that $\mathbf{E}$ and $\mathbf{F}$ are asymmetric, we can easily obtain similar results using the equality that for any $\mathbf{s}_1, \mathbf{s}_2 \in \mathbb{R}^n$ and any $\mathbf{G} \in \mathbb{R}^{n\times n}$, $\mathbf{s}_1^T \mathbf{G}\mathbf{s}_1 - \mathbf{s}_2^T \mathbf{G}\mathbf{s}_2 = 2\left(\frac{\mathbf{s}_1+\mathbf{s}_2}{2}\right)^T \mathbf{G}\left(\frac{\mathbf{s}_1-\mathbf{s}_2}{2}\right) + 2\left(\frac{\mathbf{s}_1-\mathbf{s}_2}{2}\right)^T \mathbf{G}\left(\frac{\mathbf{s}_1+\mathbf{s}_2}{2}\right)$.

where we use $\|\mathbf{F}\|_{2\to2} = O(n/m)$ and $\delta \in (0,1)$ in Eq. (53), and Eq. (54) follows from Assumption 2.4 and Eq. (46). Therefore, combining Eqs. (50) and (54), we obtain that the first term in the right-hand side of Eq. (49) can be upper bounded as follows:

$$\left|((\mathbf{v}^*)^\top \mathbf{E}\mathbf{v}^*) \cdot (\hat{\mathbf{u}} - \mathbf{v}^*)^\top \mathbf{F}(\hat{\mathbf{u}} + \mathbf{v}^*)\right|$$

$$\leq C\sqrt{\frac{k \log \frac{4Lr}{\delta}}{m}} \left( \frac{C'n\delta}{m} + C\sqrt{\frac{k \log \frac{4Lr}{\delta}}{m}} \cdot \|\bar{\mathbf{u}} - \mathbf{v}^*\|_2 \cdot \|\bar{\mathbf{u}} + \mathbf{v}^*\|_2 \right). \tag{55}$$

We have a similar bound for the second term in the right-hand side of Eq. (49), which gives

$$\left|((\mathbf{v}^*)^\top \mathbf{E}\mathbf{v}^*)(\hat{\mathbf{u}}^\top \mathbf{F}\hat{\mathbf{u}}) - (\hat{\mathbf{u}}^\top \mathbf{E}\hat{\mathbf{u}})((\mathbf{v}^*)^\top \mathbf{F}\mathbf{v}^*)\right|$$

$$\leq 2C\sqrt{\frac{k \log \frac{4Lr}{\delta}}{m}} \left( \frac{C'n\delta}{m} + C\sqrt{\frac{k \log \frac{4Lr}{\delta}}{m}} \cdot \|\bar{\mathbf{u}} - \mathbf{v}^*\|_2 \cdot \|\bar{\mathbf{u}} + \mathbf{v}^*\|_2 \right). \tag{56}$$

Moreover, for the second term in the right-hand side of Eq. (45), we have

$$(\hat{\mathbf{u}}^\top \hat{\mathbf{A}}\hat{\mathbf{u}})((\mathbf{v}^*)^\top \mathbf{F}\mathbf{v}^*) - ((\mathbf{v}^*)^\top \hat{\mathbf{A}}\mathbf{v}^*)(\hat{\mathbf{u}}^\top \mathbf{F}\hat{\mathbf{u}})$$

$$= (\hat{\mathbf{u}}^\top \hat{\mathbf{A}}\hat{\mathbf{u}})((\mathbf{v}^*)^\top \mathbf{F}\mathbf{v}^*) - ((\mathbf{v}^*)^\top \hat{\mathbf{A}}\mathbf{v}^*)((\mathbf{v}^*)^\top \mathbf{F}\mathbf{v}^*) + ((\mathbf{v}^*)^\top \hat{\mathbf{A}}\mathbf{v}^*)((\mathbf{v}^*)^\top \mathbf{F}\mathbf{v}^* - \hat{\mathbf{u}}^\top \mathbf{F}\hat{\mathbf{u}}) \tag{57}$$

$$= (\hat{\mathbf{u}} - \mathbf{v}^*)^\top \hat{\mathbf{A}}(\hat{\mathbf{u}} + \mathbf{v}^*) \cdot ((\mathbf{v}^*)^\top \mathbf{F}\mathbf{v}^*) + ((\mathbf{v}^*)^\top \hat{\mathbf{A}}\mathbf{v}^*) \cdot (\mathbf{v}^* - \hat{\mathbf{u}})^\top \mathbf{F}(\mathbf{v}^* + \hat{\mathbf{u}}), \tag{58}$$

where

$$\left|(\hat{\mathbf{u}} - \mathbf{v}^*)^\top \hat{\mathbf{A}}(\hat{\mathbf{u}} + \mathbf{v}^*)\right| = \left|(\hat{\mathbf{u}} - \bar{\mathbf{u}} + \bar{\mathbf{u}} - \mathbf{v}^*)^\top \hat{\mathbf{A}}(\hat{\mathbf{u}} - \bar{\mathbf{u}} + \bar{\mathbf{u}} + \mathbf{v}^*)\right| \tag{59}$$

$$\leq \frac{C'n\delta}{m} + \left|(\bar{\mathbf{u}} - \mathbf{v}^*)^\top \hat{\mathbf{A}}(\bar{\mathbf{u}} + \mathbf{v}^*)\right| \tag{60}$$

$$\leq \frac{C'n\delta}{m} + \|\mathbf{A}\|_{2\to2} \cdot \|\bar{\mathbf{u}} - \mathbf{v}^*\|_2 \cdot \|\bar{\mathbf{u}} + \mathbf{v}^*\|_2 + C\sqrt{\frac{k \log \frac{4Lr}{\delta}}{m}} \cdot \|\bar{\mathbf{u}} - \mathbf{v}^*\|_2 \cdot \|\bar{\mathbf{u}} + \mathbf{v}^*\|_2 \tag{61}$$

$$= \frac{C'n\delta}{m} + C_1' \cdot \|\bar{\mathbf{u}} - \mathbf{v}^*\|_2 \cdot \|\bar{\mathbf{u}} + \mathbf{v}^*\|_2, \tag{62}$$

with $C_1'$ denoting $\|\mathbf{A}\|_{2\to2} + C\sqrt{\frac{k \log \frac{4Lr}{\delta}}{m}}$. Similarly to Eq. (50), we have

$$\left|(\mathbf{v}^*)^\top \mathbf{F}\mathbf{v}^*\right| \leq C\sqrt{\frac{k \log \frac{4Lr}{\delta}}{m}}. \tag{63}$$

In addition, we have

$$\left|((\mathbf{v}^*)^\top \hat{\mathbf{A}}\mathbf{v}^*)\right| \leq \|\mathbf{A}\|_{2\to2} + C\sqrt{\frac{k \log \frac{4Lr}{\delta}}{m}} = C_1', \tag{64}$$

and

$$\left|(\mathbf{v}^* - \hat{\mathbf{u}})^\top \mathbf{F}(\mathbf{v}^* + \hat{\mathbf{u}})\right| = \left|(\mathbf{v}^* - \bar{\mathbf{u}} + \bar{\mathbf{u}} - \hat{\mathbf{u}})^\top \mathbf{F}(\mathbf{v}^* + \bar{\mathbf{u}} - \bar{\mathbf{u}} + \hat{\mathbf{u}})\right| \tag{65}$$

$$\leq \left|(\mathbf{v}^* - \bar{\mathbf{u}})^\top \mathbf{F}(\mathbf{v}^* + \bar{\mathbf{u}})\right| + \frac{C'n\delta}{m} \tag{66}$$

$$\leq C\sqrt{\frac{k \log \frac{4Lr}{\delta}}{m}} \cdot \|\bar{\mathbf{u}} - \mathbf{v}^*\|_2 \cdot \|\bar{\mathbf{u}} + \mathbf{v}^*\|_2 + \frac{C'n\delta}{m}. \tag{67}$$

Combining Eqs. (58), (62), (63), (64), (67), we obtain

$$\left|(\hat{\mathbf{u}}^\top \hat{\mathbf{A}}\hat{\mathbf{u}})((\mathbf{v}^*)^\top \mathbf{F}\mathbf{v}^*) - ((\mathbf{v}^*)^\top \hat{\mathbf{A}}\mathbf{v}^*)(\hat{\mathbf{u}}^\top \mathbf{F}\hat{\mathbf{u}})\right|$$

$$\leq 2C_1' \left( \frac{C'n\delta}{m} + C\sqrt{\frac{k \log \frac{4Lr}{\delta}}{m}} \cdot \|\bar{\mathbf{u}} - \mathbf{v}^*\|_2 \cdot \|\bar{\mathbf{u}} + \mathbf{v}^*\|_2 \right). \tag{68}$$

Similarly, we have that the third term in the right-hand side of Eq. (45) can be bounded as

$$\left| (\hat{\mathbf{u}}^\top \mathbf{E} \hat{\mathbf{u}})((\mathbf{v}^*)^\top \hat{\mathbf{B}} \mathbf{v}^*) - ((\mathbf{v}^*)^\top \mathbf{E} \mathbf{v}^*)(\hat{\mathbf{u}}^\top \hat{\mathbf{B}} \hat{\mathbf{u}}) \right|$$

$$\leq 2 C_1' \left( \frac{C' n \delta}{m} + C \sqrt{\frac{k \log \frac{4Lr}{\delta}}{m}} \cdot \|\bar{\mathbf{u}} - \mathbf{v}^*\|_2 \cdot \|\bar{\mathbf{u}} + \mathbf{v}^*\|_2 \right). \tag{69}$$

Then, combining Eqs. (45), (56), (68), (69), and setting $\delta = O(m/n)$ with a sufficiently small implied constant, we obtain

$$\left| ((\mathbf{v}^*)^\top \mathbf{A} \mathbf{v}^*)(\hat{\mathbf{u}}^\top \mathbf{B} \hat{\mathbf{u}}) - (\hat{\mathbf{u}}^\top \mathbf{A} \hat{\mathbf{u}})((\mathbf{v}^*)^\top \mathbf{B} \mathbf{v}^*) \right|$$

$$\leq 5 C_1' \left( \frac{C' n \delta}{m} + C \sqrt{\frac{k \log \frac{4Lr}{\delta}}{m}} \cdot \|\bar{\mathbf{u}} - \mathbf{v}^*\|_2 \cdot \|\bar{\mathbf{u}} + \mathbf{v}^*\|_2 \right). \tag{70}$$

Therefore, combining Eqs. (43) and (70), we obtain

$$\frac{(\mathbf{v}^*)^\top \mathbf{A} \mathbf{v}^*}{(\mathbf{v}^*)^\top \mathbf{B} \mathbf{v}^*} - \frac{\hat{\mathbf{u}}^\top \mathbf{A} \hat{\mathbf{u}}}{\hat{\mathbf{u}}^\top \mathbf{B} \hat{\mathbf{u}}} \leq \frac{5 C_1' \left( \frac{C' n \delta}{m} + C \sqrt{\frac{k \log \frac{4Lr}{\delta}}{m}} \cdot \|\bar{\mathbf{u}} - \mathbf{v}^*\|_2 \cdot \|\bar{\mathbf{u}} + \mathbf{v}^*\|_2 \right)}{d^2 \cdot \lambda_{\min}(\mathbf{B})}. \tag{71}$$

Combining Eqs. (42) and (71), we obtain

$$\sum_{i=2}^{n} g_i^2 \leq \frac{5 C_1' \cdot \lambda_{\max}(\mathbf{B}) \left( \frac{C' n \delta}{m} + C \sqrt{\frac{k \log \frac{4Lr}{\delta}}{m}} \cdot \|\bar{\mathbf{u}} - \mathbf{v}^*\|_2 \cdot \|\bar{\mathbf{u}} + \mathbf{v}^*\|_2 \right)}{d^2 (\lambda_1 - \lambda_2) \cdot \lambda_{\min}(\mathbf{B})}. \tag{72}$$

Furthermore, recall that we set $\tilde{\mathbf{u}} = \mathbf{B}^{1/2} \hat{\mathbf{u}}$ and $\tilde{\mathbf{v}}_1 = \mathbf{B}^{1/2} \mathbf{v}_1$, which leads to

$$\left\| \tilde{\mathbf{u}} - \sqrt{\sum_{i=1}^{n} g_i^2} \, \tilde{\mathbf{v}}_1 \right\|_2^2 = \|\tilde{\mathbf{u}}\|_2^2 + \sum_{i=1}^{n} g_i^2 - 2 \sqrt{\sum_{i=1}^{n} g_i^2} \, \tilde{\mathbf{u}}^\top \tilde{\mathbf{v}}_1 \tag{73}$$

$$= 2 \sum_{i=1}^{n} g_i^2 - 2 g_1 \sqrt{\sum_{i=1}^{n} g_i^2} \tag{74}$$

$$\leq 2 \sum_{i=2}^{n} g_i^2. \tag{75}$$

On the other hand, we have

$$\left\| \tilde{\mathbf{u}} - \sqrt{\sum_{i=1}^{n} g_i^2} \, \tilde{\mathbf{v}}_1 \right\|_2^2 = \left\| \mathbf{B}^{1/2} \left( \hat{\mathbf{u}} - \sqrt{\sum_{i=1}^{n} g_i^2} \, \mathbf{v}_1 \right) \right\|_2^2 \geq \lambda_{\min}(\mathbf{B}) \left\| \hat{\mathbf{u}} - \sqrt{\sum_{i=1}^{n} g_i^2} \, \mathbf{v}_1 \right\|_2^2. \tag{76}$$

Therefore, combining Eqs. (75) and (76), we obtain

$$\left\| \hat{\mathbf{u}} - \sqrt{\sum_{i=1}^{n} g_i^2} \, \mathbf{v}_1 \right\|_2^2 \leq \frac{2 \sum_{i=2}^{n} g_i^2}{\lambda_{\min}(\mathbf{B})}. \tag{77}$$

Note that by the property of projection, we have

$$\left\| \hat{\mathbf{u}} - (\hat{\mathbf{u}}^\top \mathbf{v}^*) \mathbf{v}^* \right\|_2^2 \leq \left\| \hat{\mathbf{u}} - \sqrt{\sum_{i=1}^{n} g_i^2} \, \mathbf{v}_1 \right\|_2^2. \tag{78}$$

In addition, note that for any unit vectors $\mathbf{x}_1, \mathbf{x}_2 \in \mathbb{R}^n$, if letting $\mathbf{x}_1^T \mathbf{x}_2 = \cos \alpha$, we have

$$\|\mathbf{x}_1 - (\mathbf{x}_1^\top \mathbf{x}_2) \mathbf{x}_2\|_2^2 = 1 - \cos^2 \alpha = (1 - \cos \alpha)(1 + \cos \alpha) \tag{79}$$

$$\geq \frac{1}{2} \cdot \min\{2(1 - \cos \alpha), 2(1 + \cos \alpha)\} = \frac{1}{2} \cdot \min\{\|\mathbf{x}_1 - \mathbf{x}_2\|_2^2, \|\mathbf{x}_1 + \mathbf{x}_2\|_2^2\}. \tag{80}$$

Therefore, we obtain

$$\min\left\{\|\hat{\mathbf{u}}-\mathbf{v}^*\|_2^2, \|\hat{\mathbf{u}}+\mathbf{v}^*\|_2^2\right\} \leq 2\left\|\hat{\mathbf{u}}-(\hat{\mathbf{u}}^\top\mathbf{v}^*)\mathbf{v}^*\right\|_2^2 \tag{81}$$

$$\leq \frac{4\sum_{i=2}^n g_i^2}{\lambda_{\min}(\mathbf{B})}. \tag{82}$$

Therefore, combining Eqs. (72) and (82), we obtain

$$\min\left\{\|\hat{\mathbf{u}}-\mathbf{v}^*\|_2^2, \|\hat{\mathbf{u}}+\mathbf{v}^*\|_2^2\right\}$$

$$\leq \frac{20C_1' \cdot \lambda_{\max}(\mathbf{B})\left(\frac{C'n\delta}{m} + C\sqrt{\frac{k\log\frac{4Lr}{\delta}}{m}} \cdot \|\bar{\mathbf{u}}-\mathbf{v}^*\|_2 \cdot \|\bar{\mathbf{u}}+\mathbf{v}^*\|_2\right)}{d^2(\lambda_1-\lambda_2) \cdot \lambda_{\min}^2(\mathbf{B})} \tag{83}$$

$$= C_1 \cdot \left(\frac{C'n\delta}{m} + C\sqrt{\frac{k\log\frac{4Lr}{\delta}}{m}} \cdot \|\bar{\mathbf{u}}-\mathbf{v}^*\|_2 \cdot \|\bar{\mathbf{u}}+\mathbf{v}^*\|_2\right), \tag{84}$$

where $C_1'' := \frac{20C_1' \cdot \lambda_{\max}(\mathbf{B})}{d^2(\lambda_1-\lambda_2)\cdot\lambda_{\min}^2(\mathbf{B})}$. Without loss of generality, we assume that $\hat{\mathbf{u}}^\top\bar{\mathbf{v}} \geq 0$. For this case, it follows from Eq. (84) that

$$\|\hat{\mathbf{u}}-\mathbf{v}^*\|_2^2 \leq C_1'' \cdot \left(\frac{C'n\delta}{m} + 2C\sqrt{\frac{k\log\frac{4Lr}{\delta}}{m}} \cdot \|\bar{\mathbf{u}}-\mathbf{v}^*\|_2\right) \tag{85}$$

$$\leq C_1'' \cdot \left(\frac{C'n\delta}{m} + 2C\sqrt{\frac{k\log\frac{4Lr}{\delta}}{m}} \cdot (\|\hat{\mathbf{u}}-\mathbf{v}^*\|_2 + \delta)\right). \tag{86}$$

Then, if setting $\delta = O\big((k\log\frac{4Lr}{\delta})/n\big)$ (such that $n\delta/m = O\big((k\log\frac{4Lr}{\delta})/m\big)$), we obtain

$$\|\hat{\mathbf{u}}-\mathbf{v}^*\|_2 \leq C_1 \cdot \sqrt{\frac{k\log\frac{4Lr}{\delta}}{m}}. \tag{87}$$

Similarly, if assuming $\hat{\mathbf{u}}^\top\bar{\mathbf{v}} < 0$, we obtain

$$\|\hat{\mathbf{u}}+\mathbf{v}^*\|_2 \leq C_1 \cdot \sqrt{\frac{k\log\frac{4Lr}{\delta}}{m}}, \tag{88}$$

which gives the desired result.

## C   Proof of Theorem 3.3 (Guarantees for Algorithm 1)

### C.1   Auxiliary Lemmas for Theorem 3.3

Before presenting the proof for Theorem 3.3, we first provide some useful lemmas.

**Lemma C.1.** *For any $\rho \in (\lambda_2, \lambda_1]$ and any $\mathbf{x} = \sum_{i=1}^n f_i\mathbf{v}_i \in \mathbb{R}^n$, we have*

$$(\rho-\lambda_2)\lambda_{\min}(\mathbf{B})\|\mathbf{x}\|_2^2 - (\lambda_1-\lambda_2)f_1^2 \leq \mathbf{x}^\top(\rho\mathbf{B}-\mathbf{A})\mathbf{x} \leq (\rho-\lambda_n)\lambda_{\max}(\mathbf{B})\|\mathbf{x}\|_2^2 - (\lambda_1-\lambda_n)f_1^2. \tag{89}$$

*Proof.* Let $\tilde{\mathbf{x}} = \mathbf{B}^{1/2}\mathbf{x} = \sum_{i=1}^n f_i\tilde{\mathbf{v}}_i$. We have $\|\tilde{\mathbf{x}}\|_2^2 = \sum_{i=1}^n f_i^2$ and

$$\lambda_{\min}(\mathbf{B})\|\mathbf{x}\|_2^2 \leq \sum_{i=1}^n f_i^2 = \|\tilde{\mathbf{x}}\|_2^2 = \|\mathbf{B}^{1/2}\mathbf{x}\|_2^2 \leq \lambda_{\max}(\mathbf{B})\|\mathbf{x}\|_2^2. \tag{90}$$

In addition, we have

$$\mathbf{x}^\top(\rho\mathbf{B}-\mathbf{A})\mathbf{x} = \sum_{i=1}^n f_i^2(\rho-\lambda_i) \tag{91}$$

$$= \sum_{i=2}^n (\rho-\lambda_i)f_i^2 - (\lambda_1-\rho)f_1^2. \tag{92}$$

From Eq. (92), we obtain

$$\mathbf{x}^\top(\rho\mathbf{B} - \mathbf{A})\mathbf{x} \geq (\rho - \lambda_2)\sum_{i=2}^{n} f_i^2 - (\lambda_1 - \rho)f_1^2 \tag{93}$$

$$= (\rho - \lambda_2)\sum_{i=1}^{n} f_i^2 - (\lambda_1 - \lambda_2)f_1^2 \tag{94}$$

$$\geq (\rho - \lambda_2)\lambda_{\min}(\mathbf{B})\|\mathbf{x}\|_2^2 - (\lambda_1 - \lambda_2)f_1^2. \tag{95}$$

Similarly, we have

$$\mathbf{x}^\top(\rho\mathbf{B} - \mathbf{A})\mathbf{x} \leq (\rho - \lambda_n)\sum_{i=2}^{n} f_i^2 - (\lambda_1 - \rho)f_1^2 \tag{96}$$

$$= (\rho - \lambda_n)\sum_{i=1}^{n} f_i^2 - (\lambda_1 - \lambda_n)f_1^2 \tag{97}$$

$$\leq (\rho - \lambda_n)\lambda_{\max}(\mathbf{B})\|\mathbf{x}\|_2^2 - (\lambda_1 - \lambda_n)f_1^2. \tag{98}$$

$$\square$$

**Lemma C.2.** *For any $\rho \in (\lambda_2, \lambda_1]$ and any $\mathbf{x} = \sum_{i=1}^{n} f_i\mathbf{v}_i \in \mathbb{R}^n$, $\mathbf{y} = \sum_{i=1}^{n} g_i\mathbf{v}_i \in \mathbb{R}^n$, if letting $\tau_1 = \eta(\rho - \lambda_2)\lambda_{\min}(\mathbf{B})$ and $\tau_2 = \eta(\rho - \lambda_n)\lambda_{\max}(\mathbf{B})$, we have*

$$\eta\langle(\rho\mathbf{B} - \mathbf{A})\mathbf{x}, \mathbf{y}\rangle \geq \frac{\tau_1 + \tau_2}{2}\mathbf{x}^\top\mathbf{y} - \frac{(\tau_2 - \tau_1)(\|\mathbf{x}\|_2^2 + \|\mathbf{y}\|_2^2)}{4} - \eta(\lambda_1 - \lambda_2)f_1 g_1. \tag{99}$$

*Proof.* From Lemma C.1, we have

$$\eta\langle(\rho\mathbf{B} - \mathbf{A})\mathbf{x}, \mathbf{y}\rangle = \frac{\eta(\mathbf{x} + \mathbf{y})^\top(\rho\mathbf{B} - \mathbf{A})(\mathbf{x} + \mathbf{y})}{4} - \frac{\eta(\mathbf{x} - \mathbf{y})^\top(\rho\mathbf{B} - \mathbf{A})(\mathbf{x} - \mathbf{y})}{4} \tag{100}$$

$$\geq \frac{\tau_1\|\mathbf{x} + \mathbf{y}\|_2^2}{4} - \frac{\eta(\lambda_1 - \lambda_2)(f_1 + g_1)^2}{4} - \frac{\tau_2\|\mathbf{x} - \mathbf{y}\|_2^2}{4} + \frac{\eta(\lambda_1 - \lambda_n)(f_1 - g_1)^2}{4} \tag{101}$$

$$= -\frac{(\tau_2 - \tau_1)(\|\mathbf{x}\|_2^2 + \|\mathbf{y}\|_2^2)}{4} + \frac{(\tau_1 + \tau_2)}{2}\mathbf{x}^\top\mathbf{y} - \frac{\eta(\lambda_1 - \lambda_2)(f_1 + g_1)^2}{4} + \frac{\eta(\lambda_1 - \lambda_n)(f_1 - g_1)^2}{4} \tag{102}$$

$$= -\frac{(\tau_2 - \tau_1)(\|\mathbf{x}\|_2^2 + \|\mathbf{y}\|_2^2)}{4} + \frac{(\tau_1 + \tau_2)}{2}\mathbf{x}^\top\mathbf{y} - \eta(\lambda_1 - \lambda_2)f_1 g_1 + \frac{\eta(\lambda_2 - \lambda_n)(f_1 - g_1)^2}{4} \tag{103}$$

$$\geq -\frac{(\tau_2 - \tau_1)(\|\mathbf{x}\|_2^2 + \|\mathbf{y}\|_2^2)}{4} + \frac{(\tau_1 + \tau_2)}{2}\mathbf{x}^\top\mathbf{y} - \eta(\lambda_1 - \lambda_2)f_1 g_1. \tag{104}$$

$$\square$$

**Lemma C.3.** *Suppose that $\mathbf{x} = \sum_{i=1}^{n} f_i\mathbf{v}_i$ satisfies $\|\mathbf{x}\|_2 = 1$ and $\mathbf{x}^T\mathbf{v}_1 = \nu > 0$. Let $\mathbf{h} = \mathbf{x} - \mathbf{v}^* = (f_1 - d)\mathbf{v}_1 + \sum_{i=2}^{n} f_i\mathbf{v}_i$. We have*

$$(f_1 - d)^2 \leq \left(\lambda_{\max}(\mathbf{B}) - \frac{(1 + \nu)\lambda_{\min}(\mathbf{B})}{2}\right)\|\mathbf{h}\|_2^2. \tag{105}$$

*Proof.* We have

$$(f_1 - d)^2 + \sum_{i\geq 2} f_i^2 = \left\|\mathbf{B}^{1/2}\mathbf{h}\right\|_2^2 \leq \lambda_{\max}(\mathbf{B})\|\mathbf{h}\|_2^2. \tag{106}$$

In addition, if letting $\varepsilon = 1 - \mathbf{x}^\top\mathbf{v}^* = 1 - \nu$, we have $\varepsilon < 1$ and

$$\frac{(1 + \nu)\|\mathbf{h}\|_2^2}{2} = (2 - \varepsilon)\varepsilon = \varepsilon \leq 2\varepsilon - \varepsilon^2 = \|\mathbf{x} - (1 - \varepsilon)\mathbf{v}^*\|_2^2 \leq \|\mathbf{x} - \alpha_1\mathbf{v}_1\|_2^2 \leq \frac{1}{\lambda_{\min}}\sum_{i\geq 2} f_i^2. \tag{107}$$

Combining Eqs. (106) and (107), we obtain

$$(f_1 - d)^2 \leq \left( \lambda_{\max}(\mathbf{B}) - \frac{(1 + \nu)\lambda_{\min}(\mathbf{B})}{2} \right) \|\mathbf{h}\|_2^2. \tag{108}$$

$\square$

## C.2 Complete Proof of Theorem 3.3

Let $\mathbf{h}_t = \mathbf{u}_t - \mathbf{v}^* = \mathbf{u}_t - d\mathbf{v}_1$ with $d = 1/\|\mathbf{v}_1\|_2$, and $\tilde{\mathbf{u}}_{t+1} = \mathbf{u}_t + \eta(\hat{\mathbf{A}} - \rho_t\hat{\mathbf{B}})\mathbf{u}_t$. Then, since $\mathbf{u}_{t+1} = \mathcal{P}_G(\tilde{\mathbf{u}}_{t+1})$ and $\mathbf{v}^* \in \mathcal{R}(G)$, we obtain

$$\|\tilde{\mathbf{u}}_{t+1} - \mathbf{u}_{t+1}\|_2^2 \leq \|\tilde{\mathbf{u}}_{t+1} - \mathbf{v}^*\|_2^2, \tag{109}$$

which gives

$$\|\mathbf{h}_{t+1}\|_2^2 = \|\mathbf{u}_{t+1} - \mathbf{v}^*\|_2^2 \leq 2\langle \tilde{\mathbf{u}}_{t+1} - \mathbf{v}^*, \mathbf{h}_{t+1} \rangle \tag{110}$$

$$= 2\langle \mathbf{h}_t - \eta(\rho_t\hat{\mathbf{B}} - \hat{\mathbf{A}})\mathbf{u}_t, \mathbf{h}_{t+1} \rangle \tag{111}$$

$$= 2\mathbf{h}_t^\top \mathbf{h}_{t+1} - 2\eta\langle (\lambda_1\mathbf{B} - \mathbf{A})\mathbf{u}_t, \mathbf{h}_{t+1} \rangle$$
$$\quad + 2\eta\langle (\lambda_1\mathbf{B} - \rho_t\hat{\mathbf{B}} + (\hat{\mathbf{A}} - \mathbf{A}))\mathbf{u}_t, \mathbf{h}_{t+1} \rangle \tag{112}$$

$$= 2\mathbf{h}_t^\top \mathbf{h}_{t+1} - 2\eta\langle (\lambda_1\mathbf{B} - \mathbf{A})\mathbf{h}_t, \mathbf{h}_{t+1} \rangle$$
$$\quad + 2\eta\langle (\lambda_1\mathbf{B} - \rho_t\hat{\mathbf{B}})\mathbf{u}_t, \mathbf{h}_{t+1} \rangle + 2\eta\langle (\hat{\mathbf{A}} - \mathbf{A})\mathbf{u}_t, \mathbf{h}_{t+1} \rangle \tag{113}$$

$$= 2\mathbf{h}_t^\top \mathbf{h}_{t+1} - 2\eta\langle (\lambda_1\mathbf{B} - \mathbf{A})\mathbf{h}_t, \mathbf{h}_{t+1} \rangle$$
$$\quad + 2\eta\langle (\lambda_1 - \rho_t)\mathbf{B}\mathbf{u}_t, \mathbf{h}_{t+1} \rangle + 2\eta\langle \rho_t(\mathbf{B} - \hat{\mathbf{B}})\mathbf{u}_t, \mathbf{h}_{t+1} \rangle + 2\eta\langle (\hat{\mathbf{A}} - \mathbf{A})\mathbf{u}_t, \mathbf{h}_{t+1} \rangle. \tag{114}$$

Let us bound the terms in the right-hand side of Eq. (114) separately.

- The term $\underline{2\mathbf{h}_t^\top \mathbf{h}_{t+1} - 2\eta\langle (\lambda_1\mathbf{B} - \mathbf{A})\mathbf{u}_t, \mathbf{h}_{t+1} \rangle}$: If letting

$$\mathbf{u}_t = \sum_{i=1}^n \alpha_i\mathbf{v}_i, \quad \mathbf{u}_{t+1} = \sum_{i=1}^n \beta_i\mathbf{v}_i, \tag{115}$$

we obtain

$$\mathbf{h}_t = \mathbf{u}_t - d\mathbf{v}_1 = (\alpha_1 - d)\mathbf{v}_1 + \sum_{i=2}^n \alpha_i\mathbf{v}_i, \tag{116}$$

$$\mathbf{h}_{t+1} = \mathbf{u}_{t+1} - d\mathbf{v}_1 = (\beta_1 - d)\mathbf{v}_1 + \sum_{i=2}^n \beta_i\mathbf{v}_i. \tag{117}$$

Moreover, if letting $\gamma_1 = \eta(\lambda_1 - \lambda_2)\lambda_{\min}(\mathbf{B})$ and $\gamma_2 = \eta(\lambda_1 - \lambda_n)\lambda_{\max}(\mathbf{B})$, from Lemma C.2, we obtain

$$2\mathbf{h}_t^\top \mathbf{h}_{t+1} - 2\eta\langle (\lambda_1\mathbf{B} - \mathbf{A})\mathbf{h}_t, \mathbf{h}_{t+1} \rangle \tag{118}$$

$$\leq (2 - (\gamma_1 + \gamma_2))\mathbf{h}_t^\top \mathbf{h}_{t+1} + \frac{(\gamma_2 - \gamma_1)(\|\mathbf{h}_t\|_2^2 + \|\mathbf{h}_{t+1}\|_2^2)}{2}$$
$$\quad + 2\eta(\lambda_1 - \lambda_2)(\alpha_1 - d)(\beta_1 - d) \tag{119}$$

$$\leq (2 - (\gamma_1 + \gamma_2))\mathbf{h}_t^\top \mathbf{h}_{t+1} + \frac{(\gamma_2 - \gamma_1)(\|\mathbf{h}_t\|_2^2 + \|\mathbf{h}_{t+1}\|_2^2)}{2}$$
$$\quad + \frac{2\gamma_1\|\mathbf{h}_t\|_2 \cdot \|\mathbf{h}_{t+1}\|_2}{\lambda_{\min}(\mathbf{B})}\sqrt{\lambda_{\max}(\mathbf{B}) - \frac{(1 + \nu_t)\lambda_{\min}(\mathbf{B})}{2}}\sqrt{\lambda_{\max}(\mathbf{B}) - \frac{(1 + \nu_{t+1})\lambda_{\min}(\mathbf{B})}{2}} \tag{120}$$

$$= (2 - (\gamma_1 + \gamma_2))\mathbf{h}_t^\top \mathbf{h}_{t+1} + \frac{(\gamma_2 - \gamma_1)(\|\mathbf{h}_t\|_2^2 + \|\mathbf{h}_{t+1}\|_2^2)}{2}$$
$$\quad + \gamma_1\sqrt{2\kappa(\mathbf{B}) - (1 + \nu_t)}\sqrt{2\kappa(\mathbf{B}) - (1 + \nu_{t+1})}\|\mathbf{h}_t\|_2 \cdot \|\mathbf{h}_{t+1}\|_2 \tag{121}$$

$$\leq (2 - (\gamma_1 + \gamma_2))\mathbf{h}_t^\top \mathbf{h}_{t+1} + \frac{1}{2}(\gamma_2 - \gamma_1)(\|\mathbf{h}_t\|_2^2 + \|\mathbf{h}_{t+1}\|_2^2)$$
$$\quad + \gamma_1(2\kappa(\mathbf{B}) - (1 + \nu_0))\|\mathbf{h}_t\|_2 \cdot \|\mathbf{h}_{t+1}\|_2, \tag{122}$$

where we set $\nu_t := \mathbf{u}_t^\top \mathbf{v}^*,$[7] $\kappa(\mathbf{B}) = \lambda_{\max}(\mathbf{B})/\lambda_{\min}(\mathbf{B})$, and Eq. (120) follows from Lemma C.3.

- The term $2\eta\langle(\lambda_1 - \rho_t)\mathbf{B}\mathbf{u}_t, \mathbf{h}_{t+1}\rangle$: We have

$$\lambda_1 - \rho_t = \lambda_1 - \frac{\mathbf{u}_t^\top(\mathbf{A} + \mathbf{E})\mathbf{u}_t}{\mathbf{u}_t^\top(\mathbf{B} + \mathbf{F})\mathbf{u}_t} \tag{123}$$

$$= \frac{\sum_{i=2}^n (\lambda_1 - \lambda_i)\alpha_i^2 + \mathbf{u}_t^\top(\lambda_1\mathbf{F} - \mathbf{E})\mathbf{u}_t}{\sum_{i=1}^n \alpha_i^2 + \mathbf{u}_t^\top\mathbf{F}\mathbf{u}_t}. \tag{124}$$

Note that from the calculations in Lemma C.3 (*cf.* Eq. (106)), we know

$$\sum_{i\geq 2} \alpha_i^2 \leq \lambda_{\max}(\mathbf{B})\|\mathbf{h}_t\|_2^2. \tag{125}$$

In addition, we have

$$1 = \|\mathbf{u}_t\|_2^2 = \left\|\mathbf{B}^{-1/2}\mathbf{B}^{1/2}\mathbf{u}_t\right\|_2^2 \leq \lambda_{\max}^2\left(\mathbf{B}^{-1/2}\right)\left\|\mathbf{B}^{1/2}\mathbf{u}_t\right\|_2^2 = \frac{1}{\lambda_{\min}(\mathbf{B})}\sum_{i=1}^n \alpha_i^2, \tag{126}$$

which gives

$$\sum_{i=1}^n \alpha_i^2 \geq \lambda_{\min}(\mathbf{B}). \tag{127}$$

Similarly to Eq. (47), we know that $\mathbf{u}_t$ can be written as

$$\mathbf{u}_t = \mathbf{u}_t - \bar{\mathbf{u}}_t + \bar{\mathbf{u}}_t, \tag{128}$$

where $\bar{\mathbf{u}}_t \in G(M)$ satisfies the condition that $\|\mathbf{u}_t - \bar{\mathbf{u}}_t\|_2 \leq \delta$. Then, we have

$$\left|\mathbf{u}_t^\top\mathbf{F}\mathbf{u}_t\right| = \left|(\mathbf{u}_t - \bar{\mathbf{u}}_t + \bar{\mathbf{u}}_t)^\top\mathbf{F}(\mathbf{u}_t - \bar{\mathbf{u}}_t + \bar{\mathbf{u}}_t)\right| \tag{129}$$

$$\leq \frac{C'n\delta}{m} + \left|\bar{\mathbf{u}}_t^\top\mathbf{F}\bar{\mathbf{u}}_t\right| \tag{130}$$

$$\leq \frac{C'n\delta}{m} + C\sqrt{\frac{k\log\frac{4Lr}{\delta}}{m}} \tag{131}$$

$$\leq C\sqrt{\frac{k\log\frac{4Lr}{\delta}}{m}}, \tag{132}$$

where Eq. (132) follows from the condition that $\delta = O\left((k\log\frac{4Lr}{\delta})/m\right)$. Similarly, we can obtain that

$$\left|\mathbf{u}_t^\top(\lambda_1\mathbf{F} - \mathbf{E})\mathbf{u}_t\right| \leq C(|\lambda_1| + 1)\sqrt{\frac{k\log\frac{4Lr}{\delta}}{m}}. \tag{133}$$

Combining Eqs. (124), (125), (127), (132), and (133), we obtain

$$\lambda_1 - \rho_t \leq \frac{(\lambda_1 - \lambda_n)\lambda_{\max}(\mathbf{B})\|\mathbf{h}_t\|_2^2 + C(|\lambda_1| + 1)\sqrt{\frac{k\log\frac{4Lr}{\delta}}{m}}}{\lambda_{\min}(\mathbf{B}) - C\sqrt{\frac{k\log\frac{4Lr}{\delta}}{m}}} \tag{134}$$

$$\leq \frac{3(\lambda_1 - \lambda_n)\lambda_{\max}(\mathbf{B})\|\mathbf{h}_t\|_2^2 + 3C(|\lambda_1| + 1)\sqrt{\frac{k\log\frac{4Lr}{\delta}}{m}}}{2\lambda_{\min}(\mathbf{B})}. \tag{135}$$

---

[7] By proof of induction, using the same proof strategy to follow, we can show that the sequence $\{\nu_t\}_{t\geq 0}$ is monotonically non-decreasing. We omit the details for brevity.

Therefore,

$$2\eta|\langle(\lambda_1 - \rho_t)\mathbf{B}\mathbf{u}_t, \mathbf{h}_{t+1}\rangle| \leq 2\eta(\lambda_1 - \rho_t)\lambda_{\max}(\mathbf{B})\|\mathbf{h}_{t+1}\|_2 \tag{136}$$

$$\leq 2\eta\lambda_{\max}(\mathbf{B})\|\mathbf{h}_{t+1}\|_2 \cdot \frac{3(\lambda_1 - \lambda_2)\lambda_{\max}(\mathbf{B})\|\mathbf{h}_t\|_2^2 + 3C(|\lambda_1| + 1)\sqrt{\frac{k\log\frac{4Lr}{\delta}}{m}}}{2\lambda_{\min}(\mathbf{B})} \tag{137}$$

$$= 3\eta\kappa(\mathbf{B})\|\mathbf{h}_{t+1}\|_2 \cdot \left((\lambda_1 - \lambda_n)\lambda_{\max}(\mathbf{B})\|\mathbf{h}_t\|_2^2 + C(|\lambda_1| + 1)\sqrt{\frac{k\log\frac{4Lr}{\delta}}{m}}\right) \tag{138}$$

$$\leq 3\kappa(\mathbf{B})\|\mathbf{h}_{t+1}\|_2 \cdot \left(\gamma_2\|\mathbf{h}_0\|_2 \cdot \|\mathbf{h}_t\|_2 + C\eta(|\lambda_1| + 1)\sqrt{\frac{k\log\frac{4Lr}{\delta}}{m}}\right) \tag{139}$$

$$=\leq 3\kappa(\mathbf{B})\|\mathbf{h}_{t+1}\|_2 \cdot \left(\gamma_2\sqrt{2(1 - \nu_0)} \cdot \|\mathbf{h}_t\|_2 + C\eta(|\lambda_1| + 1)\sqrt{\frac{k\log\frac{4Lr}{\delta}}{m}}\right), \tag{140}$$

where Eq. (139) follows from the setting that $\gamma_2 = \eta(\lambda_1 - \lambda_n)\lambda_{\max}(\mathbf{B})$.

- The term $2\eta\rho_t\langle(\mathbf{B} - \hat{\mathbf{B}})\mathbf{u}_t, \mathbf{h}_{t+1}\rangle$: We have

$$|\langle\mathbf{F}\mathbf{u}_t, \mathbf{h}_{t+1}\rangle| = |\langle\mathbf{F}(\mathbf{u}_t - \bar{\mathbf{u}}_t + \bar{\mathbf{u}}_t), \mathbf{u}_{t+1} - \bar{\mathbf{u}}_{t+1} + \bar{\mathbf{u}}_{t+1} - \mathbf{v}^*\rangle| \tag{141}$$

$$\leq \frac{C'n\delta}{m} + |\langle\mathbf{F}\bar{\mathbf{u}}_t, \bar{\mathbf{u}}_{t+1} - \mathbf{v}^*\rangle| \tag{142}$$

$$\leq \frac{C'n\delta}{m} + C\sqrt{\frac{k\log\frac{4Lr}{\delta}}{m}} \cdot \|\bar{\mathbf{u}}_{t+1} - \mathbf{v}^*\|_2 \tag{143}$$

$$\leq \frac{C'n\delta}{m} + C\sqrt{\frac{k\log\frac{4Lr}{\delta}}{m}} \cdot (\|\mathbf{h}_{t+1}\|_2 + \delta). \tag{144}$$

Therefore, we obtain

$$\left|2\eta\rho_t\langle(\mathbf{B} - \hat{\mathbf{B}})\mathbf{u}_t, \mathbf{h}_{t+1}\rangle\right| \leq 2\eta\lambda_1 \cdot \left(\frac{C'n\delta}{m} + C\sqrt{\frac{k\log\frac{4Lr}{\delta}}{m}} \cdot (\|\mathbf{h}_{t+1}\|_2 + \delta)\right). \tag{145}$$

- The term $2\eta\langle(\hat{\mathbf{A}} - \mathbf{A})\mathbf{u}_t, \mathbf{h}_{t+1}\rangle$: Similarly to Eq. (145), we obtain

$$\left|2\eta\langle(\hat{\mathbf{A}} - \mathbf{A})\mathbf{u}_t, \mathbf{h}_{t+1}\rangle\right| \leq 2\eta \cdot \left(\frac{C'n\delta}{m} + C\sqrt{\frac{k\log\frac{4Lr}{\delta}}{m}} \cdot (\|\mathbf{h}_{t+1}\|_2 + \delta)\right). \tag{146}$$

Combining Eqs. (114), (122), (140), (145), and (146), we obtain

$$\begin{aligned}
\|\mathbf{h}_{t+1}\|_2^2 \leq {}& (2 - (\gamma_1 + \gamma_2))\mathbf{h}_t^\top\mathbf{h}_{t+1} + \frac{\gamma_2 - \gamma_1}{2} \cdot (\|\mathbf{h}_t\|_2^2 + \|\mathbf{h}_{t+1}\|_2^2) \\
& + \gamma_1(2\kappa(\mathbf{B}) - (1 + \nu_0)) \cdot \|\mathbf{h}_t\|_2 \cdot \|\mathbf{h}_{t+1}\|_2 \\
& + 3\kappa(\mathbf{B})\|\mathbf{h}_{t+1}\|_2 \cdot \left(\gamma_2\sqrt{2(1 - \nu_0)} \cdot \|\mathbf{h}_t\|_2 + C\eta(|\lambda_1| + 1)\sqrt{\frac{k\log\frac{4Lr}{\delta}}{m}}\right) \\
& + 2\eta\lambda_1 \cdot \left(\frac{C'n\delta}{m} + C\sqrt{\frac{k\log\frac{4Lr}{\delta}}{m}} \cdot (\|\mathbf{h}_{t+1}\|_2 + \delta)\right) \\
& + 2\eta \cdot \left(\frac{C'n\delta}{m} + C\sqrt{\frac{k\log\frac{4Lr}{\delta}}{m}} \cdot (\|\mathbf{h}_{t+1}\|_2 + \delta)\right).
\end{aligned} \tag{147}$$

Then, if letting

$$c_0 = \frac{\gamma_2 - \gamma_1}{2}, \tag{148}$$

$$b_0 = (2 - (\gamma_1 + \gamma_2)) + \gamma_1(2\kappa(\mathbf{B}) - (1 + \nu_0)) + 3\gamma_2\kappa(\mathbf{B})\sqrt{2(1 - \nu_0)}, \tag{149}$$

we obtain

$$(1 - c_0)\|\mathbf{h}_{t+1}\|_2^2 \leq \left( b_0\|\mathbf{h}_t\|_2 + 2\eta(|\lambda_1| + 1)C\sqrt{\frac{k \log \frac{4Lr}{\delta}}{m}} \right) \cdot \|\mathbf{h}_{t+1}\|_2$$

$$+ c_0\|\mathbf{h}_t\|_2^2 + 2\eta(|\lambda_1| + 1)\left( \frac{C'n\delta}{m} + C\delta\sqrt{\frac{k \log \frac{4Lr}{\delta}}{m}} \right), \tag{150}$$

which leads to

$$\|\mathbf{h}_{t+1}\|_2$$

$$\leq \frac{\left( b_0\|\mathbf{h}_t\|_2 + 2\eta(|\lambda_1| + 1)C\sqrt{\frac{k \log \frac{4Lr}{\delta}}{m}} \right) + \sqrt{(1 - c_0)\left( c_0\|\mathbf{h}_t\|_2^2 + 2\eta(|\lambda_1| + 1)\left( \frac{C'n\delta}{m} + C\delta\sqrt{\frac{k \log \frac{4Lr}{\delta}}{m}} \right) \right)}}{1 - c_0} \tag{151}$$

$$\leq \frac{(b_0 + \sqrt{(1 - c_0)c_0})\|\mathbf{h}_t\|_2 + 2\eta(|\lambda_1| + 1)C\sqrt{\frac{k \log \frac{4Lr}{\delta}}{m}} + \sqrt{2\eta(1 - c_0)(|\lambda_1| + 1)\left( \frac{C'n\delta}{m} + C\delta\sqrt{\frac{k \log \frac{4Lr}{\delta}}{m}} \right)}}{1 - c_0}. \tag{152}$$

Then, if $\delta > 0$ satisfies the condition that $\delta = O\left((k \log \frac{4Lr}{\delta})/n\right)$, we obtain

$$\|\mathbf{h}_{t+1}\|_2 \leq \frac{b_0 + \sqrt{(1 - c_0)c_0}}{1 - c_0} \cdot \|\mathbf{h}_t\|_2 + C_2\sqrt{\frac{k \log \frac{4Lr}{\delta}}{m}}, \tag{153}$$

where $C_2$ is a positive constant obtained from to Eq. (152), depending on $\mathbf{A}, \mathbf{B}$, and $\eta$. This completes the proof.

## D    Additional Experiments for MNIST

In this section, we perform the experiments for the case that $\hat{\mathbf{B}}$ in Eq. (24) is modified by $\mathbf{w}_i \sim \mathcal{N}(\mathbf{0}, \text{Diag}(2, 1, \ldots, 1))$ (other settings of the experiments remain unchanged), and thus $\mathbf{B} = \mathbb{E}[\hat{\mathbf{B}}] = \text{Diag}(2, 1, \ldots, 1)$ and $\kappa(\mathbf{B}) = \lambda_{\max}(\mathbf{B})/\lambda_{\min}(\mathbf{B}) = 2$. The quantitative results are shown in the following table, from which we observe that our PRFM method also performs well in this case.

| $m$ | Rifle20 | Rifle100 | PPower | PRFM |
|---|---|---|---|---|
| 100 | $0.17 \pm 0.02$ | $0.27 \pm 0.01$ | $0.75 \pm 0.03$ | $0.80 \pm 0.02$ |
| 200 | $0.28 \pm 0.01$ | $0.43 \pm 0.01$ | $0.78 \pm 0.01$ | $0.86 \pm 0.02$ |
| 300 | $0.32 \pm 0.01$ | $0.50 \pm 0.01$ | $0.79 \pm 0.01$ | $0.91 \pm 0.01$ |

## E    Experimental Results for CelebA

The CelebA dataset contains over 200,000 face images of celebrities. Each input image is cropped for the deep convolutional generative adversarial networks (DCGAN) model with a latent dimension of $k = 100$.[8] Among 5 random restarts, we choose the best estimate. For the projection operator, the Adam optimizer is utilized with 100 steps and a learning rate of 0.1.

Similar to MNIST, we also consider two cases where $(\hat{\mathbf{A}}, \hat{\mathbf{B}})$ is generated from Eqs. (24) and (25). The experimental results are presented in Figures 3 and 4.

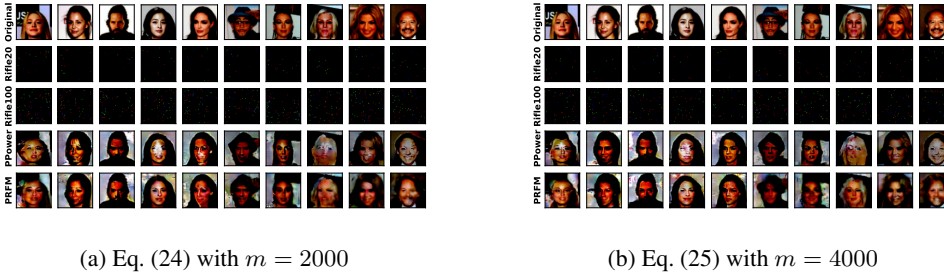

(a) Eq. (24) with $m = 2000$       (b) Eq. (25) with $m = 4000$

Figure 3: Reconstructed images of the CelebA dataset for $(\hat{\mathbf{A}}, \hat{\mathbf{B}})$ generated from Eqs. (24) and (25).

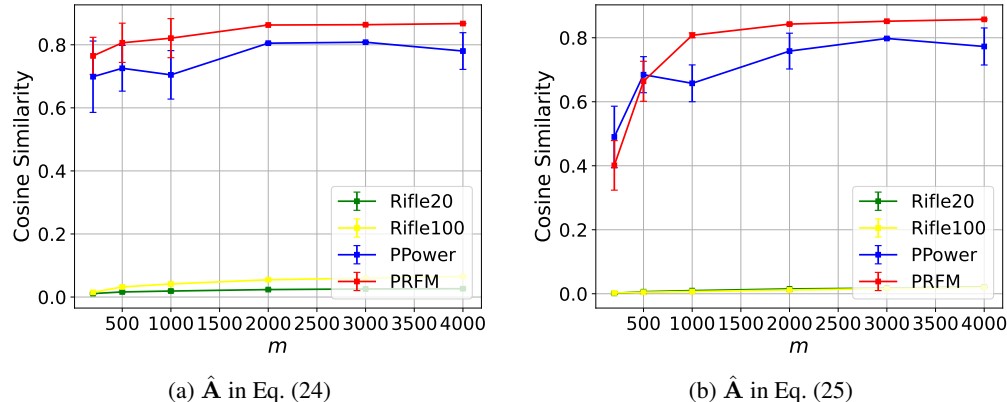

(a) $\hat{\mathbf{A}}$ in Eq. (24)       (b) $\hat{\mathbf{A}}$ in Eq. (25)

Figure 4: Quantitative results of the performance of the methods on CelebA.

While the CelebA dataset is publicly accessible and widely used, we would like to point out the potential for issues such as ethnic-group switching during face-image reconstruction.

- Bias and Stereotyping: If a generative model trained on the CelebA dataset exhibits ethnic-group switching during reconstruction, it could reinforce harmful stereotypes. For example, if certain facial features are associated with a particular ethnicity in an inaccurate or unfair way, it can lead to the misrepresentation and marginalization of that ethnic group. This can also affect how society perceives different ethnicities, potentially leading to discrimination in areas such as employment, housing, and social interactions.

- Invasion of Privacy and Consent: In the context of more sensitive data, if a generative model can manipulate or change aspects of an individual's appearance in a way that they did not consent to, it is a serious invasion of privacy. For example, if a model is used to generate images of someone in a different ethnic guise without their permission, it can cause emotional distress and harm to their self-identity. The use of such models on data where the individuals have not given proper consent for this type of manipulation can lead to legal and ethical dilemmas.

- Social and Cultural Impact: Changing the ethnicity of a face in an image can have significant social and cultural implications. It can undermine the cultural identity of individuals and communities. For instance, if a model is used to "erase" or change the ethnic features of a cultural icon in an image, it can be seen as an act of cultural appropriation or disrespect. It can also affect the way cultural heritage is represented and passed down, as the authenticity of images related to a particular culture may be compromised by unethical generative model manipulations.

---

[8]We follow the PyTorch DCGAN tutorial in `https://pytorch.org/tutorials/beginner/dcgan_faces_tutorial.html?highlight=dcgan` to pre-train the model.

