# OpenReview forum: "Generalized Eigenvalue Problems with Generative Priors"
_NeurIPS.cc/2024/Conference — NeurIPS 2024 poster_

### Official Review · Reviewer_5DuU · 2024-06-24

**Soundness:** 2
**Presentation:** 3
**Contribution:** 2
**Rating:** 5
**Confidence:** 4

**Summary:**

This paper provides theoretical guarantees for the optimal solution of the generalised eigenvalue problem with generative priors. It also designs an algorithm to approximate the optimal solution within some guaranteed distance.

**Strengths:**

The obtained results look new and rigorously proved.

**Weaknesses:**

+ Assumption 2.4 looks too strong and not very convincing. As shown in (12), this condition is even stronger than [72, Assumption 1]. At the first sight, I don't see any pair ($\mathbf{E}$,$\mathbf{F}$) which satisfies this assumption except the trivial one $\mathbf{E}=0$ and $\mathbf{F}=0$. Please find at least one class of $(\mathbf{E}, \mathbf{F})$ which satisfies this assumption.
+ Similarly, it looks not interesting to only put three conditions (21), (22), (23) in Theorem 3.3 but don't identify (at least) a class of matrices ($\mathbf{A}, \mathbf{B})$ which satisfies these conditions. Please give some classes of $(\mathbf{A}, \mathbf{B})$ such that these conditions hold. It is more interesting if $\mathbf{B}$ is not the trivial one, i.e., $\mathbf{B}=\mathbf{I}_n$, as in the experiment since you are considering the generalised eigenvalue problem.

**Questions:**

Please address my comments in the weakness. Additional notes:
+ Typo in (11): I think the constant should be $2C$ (not $\sqrt{2} C$).
+ Typo: The second and third terms in (34) and (35) should be $(\mathbf{\hat{u}}-\mathbf{w})^T \mathbf{\hat{B}}\mathbf{w}$ and $\mathbf{w}^T \mathbf{\hat{B}}(\mathbf{\hat{u}}-\mathbf{w})$.
+ Typo: The second term in (49) should be $-$.

**Limitations:**

This is a theoretical research paper, hence the negative society impact of this work is not direct. The authors mention some technical limitations of this work though assumptions.

---

> ### Author Rebuttal · Authors · 2024-08-06
>
> Thank you for your helpful feedback and comments. Our responses to the main concerns are given as follows.
>
> (**Assumption 2.4 looks too strong and not very convincing. At the first sight, I don't see any pair $(\mathbf{E}, \mathbf{F})$ which satisfies this assumption except the trivial one $\mathbf{E} = \mathbf{0}$ and $\mathbf{F} = \mathbf{0}$. Please find at least one class of $(\mathbf{E}, \mathbf{F})$  which satisfies this assumption**) As stated in the paragraph preceding Assumption 2.4, similarly to the proof for the spiked covariance model in [45, Appendix B.1], it can be readily demonstrated that for the $(\hat{\mathbf{A}}, \hat{\mathbf{B}})$ constructed in Section 4.2, the corresponding perturbation matrices $\mathbf{E}$ and $\mathbf{F}$ satisfy Assumption 2.4 with high probability. More specifically, for Eq. (29) with $$\hat{\mathbf{A}} = \frac{1}{m}\sum_{i=1}^m(2\gamma_i\mathbf{v}^*+\mathbf{z}_i)(2\gamma_i\mathbf{v}^*+\mathbf{z}_i)^\top,$$
>
> and $\hat{\mathbf{B}} =\frac{1}{m}\sum_{i=1}^m \mathbf{w}_i\mathbf{w}_i^\top$, we have $\mathbb{E}[\hat{\mathbf{A}}] = 4\mathbf{v}^*(\mathbf{v}^*)^\top + \mathbf{I}_n$ and $\mathbb{E}[\hat{\mathbf{B}}] = \mathbf{I}_n$;
>
> where $\mathbf{v}^*$ is a unit vector; $\gamma_i \sim \mathcal{N}(0,1)$, $\mathbf{z}_i \sim  \mathcal{N}(\mathbf{0},\mathbf{I}_n)$, $\mathbf{w}_i\sim  \mathcal{N}(\mathbf{0},\mathbf{I}_n)$, and they are mutually independent. If setting $\mathbf{A} = \mathbb{E}[\hat{\mathbf{A}}]$ and $\mathbf{E} = \hat{\mathbf{A}} - \mathbf{A} \neq \mathbf{0}$, and setting $\mathbf{B} = \mathbb{E}[\hat{\mathbf{B}}]$ and $\mathbf{F} = \hat{\mathbf{B}} - \mathbf{B} \neq \mathbf{0}$, we obtain for **any** $\mathbf{s}_1 \in S_1$, $\mathbf{s}_2 \in S_2$,
>
> $$\left|\mathbf{s}_1^\top \mathbf{E} \mathbf{s}_2\right|$$
>
> $$ = \left|\frac{1}{m}\sum_{i=1}^m \left(\mathbf{s}_1^\top(2\gamma_i\mathbf{v}^*+\mathbf{z}_i) \cdot \mathbf{s}_2^\top(2\gamma_i\mathbf{v}^*+\mathbf{z}_i) - \mathbb{E}\left[\mathbf{s}_1^\top(2\gamma_i\mathbf{v}^*+\mathbf{z}_i) \cdot\mathbf{s}_2^\top(2\gamma_i\mathbf{v}^*+\mathbf{z}_i)\right]\right)\right|.$$
>
> Since $\mathbf{s}_1^\top(2\gamma_i\mathbf{v}^*+\mathbf{z}_i) \mathbf{s}_2^\top(2\gamma_i\mathbf{v}^*+\mathbf{z}_i)$ are sub-exponential and independent (with the sub-exponential norm being upper bounded by $C\\|\mathbf{s}_1\\|_2\cdot \\|\mathbf{s}_2\\|_2$, where $C$ is an absolute constant), using the concentration inequality for the sum of independent sub-exponential random variables [75, Proposition 5.16], we obtain for any $u > 0$ satisfying $m = \Omega(u)$, the following holds with probability $1-e^{-\Omega(u)}$:
>
> $$\left|\mathbf{s}_1^\top \mathbf{E} \mathbf{s}_2\right| \le  C \\|\mathbf{s}_1\\|_2\cdot \\|\mathbf{s}_2\\|_2 \cdot\sqrt{\frac{u}{m}}.$$
>
> Taking a union bound for **all** $\mathbf{s}_1 \in S_1$, $\mathbf{s}_2 \in S_2$, and setting $u = \log (|S_1|\cdot |S_2|)$, we obtain with  probability $1-e^{-\Omega(\log (|S_1|\cdot |S_2|))}$ that the following holds for all **all** $\mathbf{s}_1 \in S_1$, $\mathbf{s}_2 \in S_2$:
>
>  $$\left|\mathbf{s}_1^\top \mathbf{E} \mathbf{s}_2\right| \le  C \\|\mathbf{s}_1\\|_2\cdot \\|\mathbf{s}_2\\|_2 \cdot \sqrt{\frac{\log (|S_1|\cdot |S_2|)}{m}}.$$
>
> Please refer to [45, Appendix B.1] for more technical details. Similarly, for $\big|\mathbf{s}_1^\top \mathbf{F} \mathbf{s}_2\big|$, we can obtain the upper bound in Eq. (7).
>
> (**It looks not interesting to only put three conditions (21), (22), (23) in Theorem 3.3 but don't identify (at least) a class of matrices $(\mathbf{A}, \mathbf{B})$ which satisfies these conditions.  It is more interesting if $ \mathbf{B}$ is not the trivial one, i.e., $\mathbf{B}=\mathbf{I}_n$, as in the experiment since you are considering the generalised eigenvalue problem**) We have deliberated in Remark 3.4 regarding when the three conditions are satisfied. Specifically, we demonstrate that when $\kappa(\mathbf{B}) = \lambda_{\max}(\mathbf{B})/\lambda_{\min}(\mathbf{B})$ is close to 1 (or more precisely, $\kappa(\mathbf{B})<c’$ for some constant $c’>1$; note that we do not make any attempt to refine the constant $c'$ here, and we believe that our theorem actually holds under a more relaxed condition), the three conditions can be satisfied. Since this work is primarily theoretical, we follow [8] to only conduct the experiments for the case where $\mathbf{B}=\mathbf{I}_n$. We have added the experiments for the case that $\hat{\mathbf{B}}$ in Eq. (29) is modified by $\mathbf{w}_i \sim \mathcal{N}(\mathbf{0}, \mathrm{Diag}(2,1,\ldots,1))$ (other settings of the experiments remain unchanged), and thus $\mathbf{B} = \mathbb{E}[\hat{\mathbf{B}}] = \mathrm{Diag}(2,1,\ldots,1)$ and $\kappa(\mathbf{B}) = 2$. The quantitative results are as follows:
>
> | $m$| Rifle20 | Rifle100 | PPower | PRFM |
> | :---------------- | :------: | ----: | :------: | ----: |
> | 100  |   0.17 $\pm$ 0.02   | 0.27 $\pm$ 0.01 | 0.75 $\pm$ 0.03   | 0.80 $\pm$ 0.02 |
> | 200 |  0.28 $\pm$ 0.01   | 0.43 $\pm$ 0.01 | 0.78 $\pm$ 0.01   | 0.86 $\pm$ 0.02 |
> | 300 |  0.32 $\pm$ 0.01   | 0.50 $\pm$ 0.01 | 0.79 $\pm$ 0.01   | 0.91 $\pm$ 0.01 |
>
> From the above results, we observe that our PRFM method also performs well in this case. We will incorporate the corresponding numerical results in the revised version.
>
> (**Typos**) Thanks for pointing out these typos. In the revised version, we will correct them and meticulously check the manuscript to prevent typos.

---

> > ### Author Response · Authors · 2024-08-13
> >
> > Dear Reviewer 5DuU,
> >
> > We have taken your initial feedback into meticulous consideration in our responses. Could you please check whether our responses have appropriately addressed your concerns? If so, could you please kindly consider increasing your initial score accordingly? Certainly, we are more than happy to answer your further questions.
> >
> > Thank you for your valuable time and dedicated effort in reviewing our work!
> >
> > Best Regards,
> >
> > Authors

---

> > > ### Comment · Reviewer_5DuU · 2024-08-14
> > > **Reply to the author rebuttal**
> > >
> > > Thank you very much for your detailed answers to my comments. Based on your answers, I raised my score to 5. Please add these to your revised version.

---

### Official Review · Reviewer_cuQm · 2024-07-06

**Soundness:** 4
**Presentation:** 3
**Contribution:** 3
**Rating:** 7
**Confidence:** 4

**Summary:**

The paper studies generalized eigenvalue problems under generative priors. They show that under suitable conditions on the prior assumptions on the perturbation matrices, the optimal solution vector of the corresponding optimization problem attains the statistically optimal rate. Furthermore, they provide an algorithm that under assumptions on the signal strength, step size, and initialization vector, converges linearly to a statistically optimal solution vector. They further supplement their theoretical analysis with experiments on the MNIST dataset, while comparing with other approaches for solving generative generalized eigenvalue problems.

**Strengths:**

- The paper generalizes analysis of PCA, Fisher's Discriminant Analysis, and Canonical Correlation Analysis with generative priors and presents a unified view.
- The optimization problem (6) and the corresponding Algorithm (1), seem to attain the statistically optimal rate under certain assumptions on the perturbation matrices, signal strength, step-sizes, and smoothness of the generative priors.
- The numerical experiments complement their theoretical results and show a clear empirical improvement over existing methods.

**Weaknesses:**

- Regarding Computational Efficiency: As acknowledged by the authors, projection on the prior set may not be efficiently achievable in general and approximations may be required. Theoretical analysis taking the approximation into account would be good to see. Furthermore, even with the current approximate projection based off gradient descent, I would like to see how the proposed algorithm compares in compute time to other methods such as PPower.

- Regarding Eq (22): It seems to me that the condition assumed here essentially implies a local convergence guarantee. In Line 252, the authors mention that if $\nu_{0}$ is close to 1, then the conditions (21), (22) are easy to verify. However, achieving $\nu_{0}$ close to 1 itself seems like a hard problem since it would imply that you already have a pretty good initialization to start with, which may be hard to find in high dimensions.

**Questions:**

- Regarding practical examples/instances:  Apart from the numerical experiments pointed out in the paper, are there other scenarios where assuming existence of such generative priors is reasonable/well-accepted?

- Regarding Assumption 2.4. : Outside the spiked covariance model, can the authors specify other covariance matrices where this assumption is satisfied?

**Limitations:**

The primary limitation seems to be assuming the existence of an exact projection step. The authors address this adequately.

---

> ### Author Rebuttal · Authors · 2024-08-06
>
> Thank you for your recognition of this paper as well as the beneficial comments and questions. Our responses to the major concerns are as follows.
>
> (**Theoretical analysis taking the approximation into account would be good to see**) Similar to previous works such as [45, 59, 65], we assume exact projection in the theoretical analysis. We agree that theoretical analysis taking the approximation into account is an intriguing direction and we leave it for future research.
>
> (**Even with the current approximate projection based off gradient descent, I would like to see how the proposed algorithm compares in compute time to other methods such as PPower**)  The projection step onto the range of the generative model occupies the majority of the running time of the algorithm. Consequently, the compute time of PRFM is approximately the same as PPower, with only a minor increase due to more matrix-vector multiplications in each iteration. In the following table, we present the running time (in seconds) for PRFM and PPower on MNIST with $\hat{\mathbf{A}}$ and $\hat{\mathbf{B}}$ generated from Eq. (29) (and the number of iterations for both algorithms is set to 10; the running time is averaged over 10 test images and 10 restarts). Note that we run the algorithms given $\hat{\mathbf{A}}, \hat{\mathbf{B}} \in \mathbb{R}^{n\times n}$ (or $\mathbf{V} \in \mathbb{R}^{n\times n}$ for PPower), and the running time will not increase with $m$.
>
> | $m$ | PPower | PRFM |
> | :---------------- | :------: | ----: |
> | 100   | 38.64 $\pm$ 0.09   | 39.15 $\pm$ 0.35 |
> | 200 |  38.59 $\pm$ 0.09   | 39.25 $\pm$ 0.31 |
> | 300 |  38.61 $\pm$ 0.15   | 39.03 $\pm$ 0.33 |
>
> (**Achieving $\nu_0$ close to 1 itself seems like a hard problem since it would imply that you already have a pretty good initialization to start with, which may be hard to find in high dimensions**) Thanks for the comment. While this initialization condition might seem restrictive, it seems to be the mildest initialization condition that we can assume for the case of generative priors. The reason is as follows: For sparse priors, the initialization vector can be obtained via a convex program (see [72]). Regrettably, for a generative model, this idea no longer applies since, without additional assumptions, the problem cannot be relaxed to a convex optimization problem. We observe that similar initialization conditions have been imposed in previous works such as [31, 45]. Additionally, in practice, we do not enforce such an initialization condition to hold and we discover that simply setting the initial vector to be $[1,1,\ldots,1]^\top/\sqrt{n} \in \mathbb{R}^n$ in the numerical simulations performs well.
>
> (**Apart from the numerical experiments pointed out in the paper, are there other scenarios where assuming the existence of such generative priors is reasonable/well-accepted**) From the successful applications of deep generative models in diverse fields, there are various scenarios where assuming the existence of such generative priors is reasonable and well-accepted. For instance, robust compressed sensing MRI with generative priors has been investigated in
>
> Jalal, A., Arvinte, M., Daras, G., Price, E., Dimakis, A.G. and Tamir, J., 2021. Robust compressed sensing MRI with deep generative priors. Advances in Neural Information Processing Systems, 34, pp.14938-14954.
>
> Additionally, an approach similar to the closely related PPower method has been applied to Interferometric Passive Radar Imaging in
>
> Kazemi, S., Yonel, B. and Yazici, B., 2023. Interferometric Passive Radar Imaging With Deep Denoising Priors. IEEE Transactions on Aerospace and Electronic Systems, 60(1), pp.145-156.
>
> For more real applications of deep generative models in inverse problems in imaging, please refer to the following popular survey paper [569 citations]:
>
> Ongie, G., Jalal, A., Metzler, C.A., Baraniuk, R.G., Dimakis, A.G. and Willett, R., 2020. Deep learning techniques for inverse problems in imaging. IEEE Journal on Selected Areas in Information Theory, 1(1), pp.39-56.
>
> (**Outside the spiked covariance model, can the authors specify other covariance matrices where Assumption 2.4 is satisfied**) Assumption 2.4 is also satisfied with high probability by the matrices corresponding to the phase retrieval model (see Eq. (30)). Additionally, similar to what is mentioned in [72], it is straightforward to verify that under generative priors, for various statistical models that can be formulated as a GEP such as CCA and FDA, Assumption 2.4 will be satisfied with high probability.

---

> > ### Comment · Reviewer_cuQm · 2024-08-11
> >
> > I thank the authors for their response. I maintain my current score.

---

### Official Review · Reviewer_PACz · 2024-07-16

**Soundness:** 2
**Presentation:** 2
**Contribution:** 2
**Rating:** 5
**Confidence:** 3

**Summary:**

This submission proposes a way to solve Generalized Eigenproblems with a constraint on the eigenvectors to be in the range of some generative model. On top of a simple algorithm, the paper proposes bound on the optimal solution and some experiments on some toy data.

**Strengths:**

Generalized eigenproblems are a cornerstone of many ML problems and their study could have a great impact.
The idea of constraining the solution with the output of a generative model (although a bit unclear) has a good potential.

**Weaknesses:**

The presentation of the problem could be improved (for exemple, it is never said that Eq 3 is a Rayleigh quotient), section 2.2 gather folklore results that are not useful (in my opinion).
It is also difficult to follow how the algorithm 1 is derived from section 3.
The numerical experiments are carried out on toy data and it makes it difficult to understand how it could be used in practice.

**Questions:**

I am not quite sure to understand how the generative prior (and the projection onto the range of a pre trained model) is not redondant with the optimization problem itself. If the data are distributed according to some distribution, then the covariance of the data should be in this range. There is for sure something that I missed and it was not clear from the application but could you clarify this point ?

In Th 3.3, beside being a technical condition, is there any impact of the condition \gamma_1 + \gamma_2 < 2 ? Is it a reasonable constraint ?

In the main results, the $\mathcal{P}_G(\dot)$ has to be exact. In the experiments, it is approximated. How does this approximation impact the algorithm and the numerical results ?

The proposed algorithm is somewhat similar to a projected power method. How stable is it when k grows ? (As it is know to behave poorly as k grows).
On possible solution could be to solve the problem as the $\max_U Tr (U^T A U)$ with U in a genaralized Stiefel manifold (defined with B) and the generative prior could be enforced with a regularization. This should prove to be more stable than the power method-like method.

Section C of the appendix is very interesting and should be included in the paper (although the argument about having a proof that is bigger than the competitors is clearly not a valid argument …)

**Limitations:**

Since the paper involves the use of a generative model, it could have some negative implication for the society. A discussion on that topic should be included.

---

> ### Author Rebuttal · Authors · 2024-08-06
>
> Thank you for your helpful feedback and suggestions. Our responses to the main concerns are given as follows.
>
> (**The presentation of the problem could be improved**) Thank you for the comments. We will undertake revisions on the presentation, encompassing the following aspects: 1) Mention that Eq. (3) is a Rayleigh quotient; 2) Relocate Section 2.2 to the Appendix; 3) Incorporate more explanations for the derivation of Algorithm 1; 4) Include Section C of the Appendix in the main paper (and we will meticulously modify it to prevent overstated claims).
>
> (**The numerical experiments are carried out on toy data and it makes it difficult to understand how it could be used in practice**) We have also conducted the experiments for the CelebA dataset (which comprises over 200,000 face images of celebrities and the image vector has a dimension of $n = 64 \times 64  \times 3 = 12288$) in the case where $\hat{\mathbf{A}}$ and $\hat{\mathbf{B}}$ are generated from Eq. (29), and the quantitative results (in terms of Cosine Similarity) are as follows.
>
> | $m$ | PPower | PRFM |
> | :---------------- | :------: | ----: |
> | 200   | 0.70 $\pm$ 0.11   | 0.76 $\pm$ 0.06 |
> | 1000 |  0.73 $\pm$ 0.07   | 0.81 $\pm$ 0.06 |
> | 3000 |  0.78 $\pm$ 0.04   | 0.87 $\pm$ 0.01 |
>
> We will incorporate the experimental results on the CelebA dataset in the revised version.
>
> (**I am not quite sure to understand how the generative prior (and the projection onto the range of a pre-trained model) is not redundant with the optimization problem itself**) The generative prior is not redundant with the optimization problem. The rationale is as follows: We concentrate on the high-dimensional scenario where the number of samples $m$ is significantly smaller than the data dimension $n$. To attain reliable reconstruction in this context, it is necessary to impose a low-dimensional prior on the underlying signal and add the corresponding constraint (or regularization) to the optimization problem. A common selection for the data prior is the sparse prior (note that for the popular sparse PCA/GEP problem, the constraint corresponding to the sparse prior also needs to be imposed, as seen in Eq. (5)). Given the numerous successful applications of deep generative models, we are aware that generative priors can be far more potent in modeling the distribution of real data. Hence, we follow [4] that considers generative priors for the underlying signal, and impose the corresponding constraint into the optimization problem.
>
> (**Is there any impact of the condition $\gamma_1 + \gamma_2 < 2$ ? Is it a reasonable constraint?**) The condition $\gamma_1 + \gamma_2 < 2$ can be expressed as $\eta (\lambda_1 -\lambda_2)\lambda_{\min}(\mathbf{B}) + \eta (\lambda_1 -\lambda_n)\lambda_{\max}(\mathbf{B}) < 2$, where $\lambda_1 > \lambda_2 \ge \ldots \lambda_n$ are the generalized eigenvalues of $(\mathbf{A}, \mathbf{B})$. This is a reasonable constraint that is similar to the constraint $\eta \lambda_{\max}(\mathbf{B}) < 1/(1+c)$ in  [72, Theorem 1] (where $c > 0$ is a constant specified in [72, Assumption 1]). More precisely, note that $\eta/\rho_{t-1}$ in [72, Algorithm 1] plays the role as our $\eta$ and $\rho_{t-1} \approx \lambda_1$. Then, if employing our notation, the constraint in [72, Theorem 1] is approximately $\eta \lambda_1 \lambda_{\max}(\mathbf{B}) < 1/(1+c)$, whose impact is to impose an upper bound on the selection of the step size $\eta$.
>
> (**How does the approximation of the projection step impact the algorithm and the numerical results?**) Since the projection step cannot be exact in the experiments, we adhere to previous works such as [45, 59, 65] and employ a gradient descent method along with the Adam optimizer to approximately carry out the projection step. From the experimental results, we observe that the approximation of the projection step proves effective as we are able to obtain reasonably good reconstructions when the number of samples $m$ is significantly smaller than the data dimension $n$.
>
> (**How stable is it when $k$ grows ? (As it is known to behave poorly as $k$ grows). One possible solution could be to solve the problem as the $\max_{\mathbf{U}}\mathrm{Tr}(\mathbf{U}^\top\mathbf{A}\mathbf{U})$ with $\mathbf{U}$ in a generalized Stiefel manifold (defined with $\mathbf{B}$) and the generative prior could be enforced with a regularization**) In our experiments, we employ a pre-trained generative model with the latent dimension $k$ remaining fixed. For instance, for the generative model pre-trained for the MNIST dataset, $k = 20$, and for the generative model pre-trained for the CelebA dataset, $k = 100$. We do not modify the generative model (and the latent dimension) as pre-training is time-consuming. Nevertheless, we are grateful to the reviewer for highlighting this intriguing optimization problem to us and we concur that it constitutes a promising direction for further investigation.
>
> (**Limitations and Ethics Review**) Similar to the series of works following [4], we only conduct the experiments and pre-train the generative models for the publicly accessible and widely utilized datasets MNIST and CelebA. We believe that there should be no negative implication for the society and no requirement for an ethics review.

---

> ### Comment · Reviewer_PACz · 2024-08-12
>
> I thanks the authors for their answer which covers some of my concerns and I will raise my score.
>
> About the limitations and ethics, although the authors only work on pre-trained models with classical dataset, they are tinkering with generative models. The implications for other generative models (on more sensitive data/applications) could be huge and it would deserve some discussion.

---

> > ### Author Response · Authors · 2024-08-13
> > **Responses to Reviewer PACz**
> >
> > Thank you for your responses and for raising the score. We will incorporate a discussion concerning the application of generative models to more sensitive data/applications in the revised version.

---

### Decision · Program_Chairs · 2024-09-25

**Decision:**

Accept (poster)

**Comment:**

This manuscript address how to solve symmetric (semi-definite) generalized eigenvalue problems when there is a reasonable generative prior that can be placed on the eigenvectors. Theoretical results are provided that highlight the quality of the optimal solution and practical algorithms are discussed.

Overall, the reviewers feel that there is a clear contribution here given both the prevalence of GEPs and the advent of using generative models as priors. While the algorithm is similar in spirit to prior work, the theoretical discussions highlight its potential efficacy.

Nevertheless, there are some lingering concerns about the validity/sensibility of various assumptions and the limited computational experiments. The authors rebuttal did partially address the former point, though I do think that there are som lingering questions. Of note, some of the assumptions (such as an exact projection) do appear elsewhere and are not unreasonable to make when analyzing the problem—plus the limitation of this assumption is clearly discussed/addressed.

NB: one aspect of the presentation that could use some work is a lack of engagement with the broader symmetric GEP literature. For example, deflation (as alluded to in lines 25 and 26) is not commonly preferred for targeting multiple eigenvalue/vector pairs and many other algorithms exist (such as generalizations of Rayleigh quotient iteration, subspace style iterations, the QZ method, etc.). I would encourage the authors to provide some pointers to the broader literature for solving GEPs since that likely is quite relevant here. In addition, the authors should be more clear that their results are for symmetric semi-definite GEPs—this is not always clear. Admittedly the use of SGEP for sparse GEPs complicates this matter from a notational perspective, but the results are very clearly limited to the symmetric case and that should be more clear (e.g., in the title and abstract). There are also lots of works on more general (i.e., not symmetric) GEPs, but this work does not fall in that category.